# GIFT: Bootstrapping Image-to-CAD Program Synthesis via Geometric Feedback

**Giorgio Giannone** [1 2]  **Anna Clare Doris** [2]  **Amin Heyrani Nobari** [2]  **Kai Xu** [1]  **Akash Srivastava** [3 4]  **Faez Ahmed** [2]

## Abstract

Generating executable CAD programs from images requires alignment between visual geometry and symbolic program representations, a capability that current methods fail to learn reliably as design complexity increases. Existing fine-tuning approaches rely on either limited supervised datasets or expensive post-training pipelines, resulting in brittle systems that restrict progress in generative CAD design. We argue that the primary bottleneck lies not in model or algorithmic capacity, but in the scarcity of diverse training examples that align visual geometry with program syntax. This limitation is especially acute because the collection of diverse and verified engineering datasets is both expensive and difficult to scale, constraining the development of robust generative CAD models. We introduce *Geometric Inference Feedback Tuning* (GIFT), a data augmentation framework that leverages geometric feedback to turn test-time compute into a bootstrapped set of high-quality training samples. GIFT combines two mechanisms: *Soft-Rejection Sampling* (GIFT-REJECT), which retains diverse high-fidelity programs beyond exact ground-truth matches, and *Failure-Driven Augmentation* (GIFT-FAIL), which converts near-miss predictions into synthetic training examples that improve robustness on challenging geometries. By amortizing inference-time search into the model parameters, GIFT captures the benefits of test-time scaling while reducing inference compute by 80%. It improves mean IoU by 12% over a strong supervised baseline and remains competitive with more complex multimodal systems, without requiring additional human annotation or specialized architectures.

---

[1]AI Innovation, Red Hat [2]DeCoDE Lab, MIT [3]Core AI, IBM [4]MIT-IBM Watson AI Lab. Correspondence to: <ggiorgio@mit.edu>.

*Proceedings of the 43rd International Conference on Machine Learning*, Seoul, South Korea. PMLR 306, 2026. Copyright 2026 by the author(s).

## 1. Introduction

From its theoretical origins (Ross &Rodriguez, 1963; Coons, 1963; Mantyla, 1988), Computer-Aided Design (CAD) has evolved into the cornerstone of modern engineering, enabled by the sophisticated modeling capabilities of geometric kernels (Piegl &Tiller, 2012).

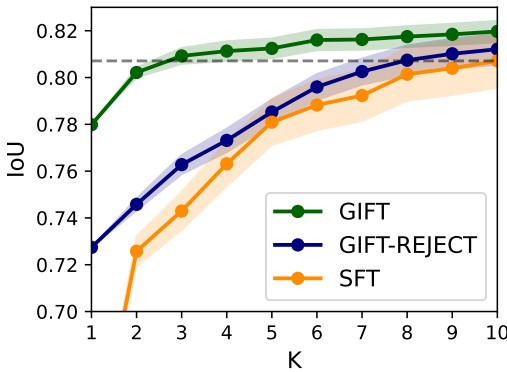

*Figure 1.* Efficiency vs. Performance. We compare Pass@k (test set IoU) across compute budgets. GIFT (green) matches the peak performance of the CAD-Coder-SFT baseline (orange) while using 80% less compute (requiring far fewer samples). GIFT outperforms both SFT and GIFT-REJECT at every compute level, demonstrating that self-training with geometric feedback significantly enhances image-conditional CAD generation. Results reflect mean IoU on the GenCAD test subset, including failure cases. An extended scaling analysis is provided in Appendix C (Figure 12).

Deep generative modeling is transforming engineering design, offering new solutions to complex, high-dimensional problems (Ahmed et al., 2025; Regenwetter et al., 2022; Song et al., 2023). Recent empirical successes highlight the versatility of these methods in tasks such as topology optimization (Nie et al., 2021; Mazé &Ahmed, 2023; Giannone et al., 2023; Nobari et al., 2025), mechanism linkage design (Nobari et al., 2024), vehicle dynamics (Elrefaie et al., 2024), and CAD generation (Alam &Ahmed, 2024).

While researchers have developed direct forward models to synthesize 3D shapes from diverse inputs like point clouds, images, and text (Alam &Ahmed, 2024; Yu et al., 2025; Tsuji et al., 2025), these approaches face a critical bottleneck: the output format. Existing generative models (Lambourne et al., 2021; Li et al., 2025b) typically yield tessel-

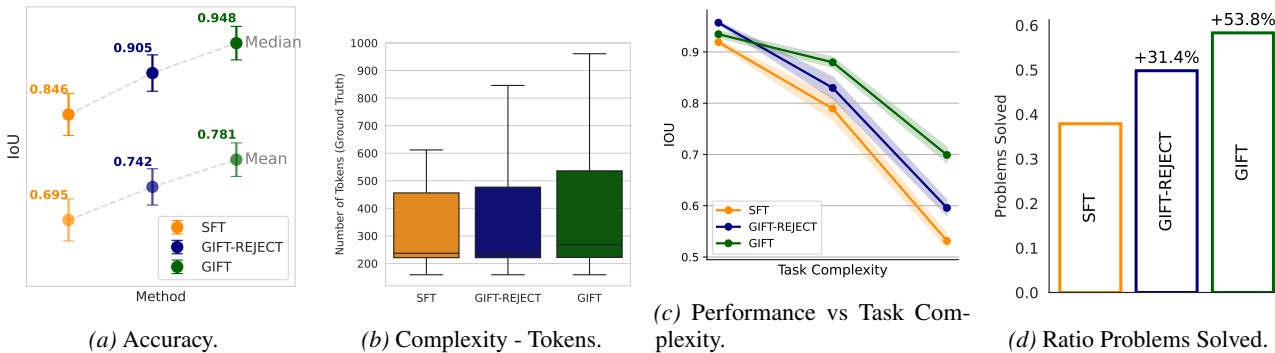

*(a)* Accuracy.     *(b)* Complexity - Tokens.     *(c)* Performance vs Task Complexity.     *(d)* Ratio Problems Solved.

*Figure 2.* Robustness Analysis. (*a*) GIFT achieves superior accuracy (Mean/Median IoU). (*b, c*) While all models degrade with increased task complexity (token length), GIFT maintains higher resilience than SFT. (*d*) GIFT solves 53% more problems compared to the baseline, highlighting the benefit of diverse training data.

lated meshes (STL) or Boundary Representations (B-Rep, STEP). These formats are inherently verbose and topologically complex, making them difficult for engineers to parameterize or edit. Moreover, the need for specialized 3D decoders or large language models to generate such structures imposes significant computational and structural overhead.

**Modality-to-Program** To circumvent these limitations, a growing paradigm leverages high-level symbolic programs (Bhuyan et al., 2024; Hewitt et al., 2020; Jones et al., 2022) as an intermediate representation. Unlike rigid geometric formats, code offers a compact and editable structure (Ganeshan et al., 2023) that is naturally amenable to autoregressive language modeling - a property that has already proven effective in domains such as mathematical reasoning (Guo et al., 2025) and visual-language grounding (Surís et al., 2023). This approach effectively abstracts geometry into parametric operations, though it necessitates an execution environment (e.g., a geometric kernel like OpenCASCADE [1]) to render the generated code into valid CAD formats. Despite this dependency, the modality-to-program strategy has gained significant traction in recent literature (Doris et al., 2025; Kolodiazhnyi et al., 2025; Li et al., 2025c; Rukhovich et al., 2025; Wang et al., 2025b), with a particular emphasis on `CadQuery` [2], a Python-based scripting language for parametric CAD design.

**Post-Training** The dominant paradigms for post-training (Ouyang et al., 2022) are Supervised Fine-Tuning (SFT) and Reinforcement Learning (RL). SFT provides a foundation for CAD generation but often suffers from weak alignment between visual features and program syntax. Recent works attempt to mitigate this by conditioning on auxiliary modalities such as point clouds and depth maps, though reliance on ad-hoc architectures limits model generality (Wang et al., 2025b; Chen et al.,

2025). RL can improve alignment and performance but is resource-intensive (Schulman et al., 2017; Guo et al., 2025), incurring high memory and communication costs due to the need to execute CPU-bound CAD kernels during online training (Li et al., 2025c; Kolodiazhnyi et al., 2025).

We argue, however, that the primary bottleneck in SFT and RL post-training for engineering design is not model quality or algorithmic sophistication, but the fine-tuning data itself. Our approach is motivated by a simple observation: advances in LLMs and VLMs have been driven largely by high-quality, diverse datasets, yet assembling such data for data-driven engineering design remains expensive. Data augmentation offers a natural remedy, but the central challenge is determining how to use available compute most effectively to address model strengths and weaknesses while exploiting the structural properties of the underlying engineering problem.

To address these limitations, we introduce *Geometric Inference Feedback Tuning* (GIFT), a framework for verifier-guided data augmentation in Image-to-CAD generation. GIFT uses offline geometric verification to mine diverse valid programs and structured near-miss failures, converting them into a high-quality augmented training set. By distilling the benefits of test-time search into the training data, our approach improves generation quality while reducing inference cost and avoiding the complexity of online reinforcement learning.

GIFT separates exploration from learning: candidate programs are generated and verified offline during inference-time sampling, then used for standard supervised updates. This design preserves the benefits of reinforced geometric feedback without introducing the instability of online training.

**Contributions** Our key contributions are:

- We introduce *Geometric Inference Feedback Tuning*

---

[1] https://github.com/Open-Cascade-SAS/OCCT
[2] https://github.com/CadQuery/cadquery

(GIFT), a scalable data augmentation framework for Image-to-CAD program synthesis that uses offline geometric feedback to convert inference-time samples into augmented supervised training data.

- We propose a dual augmentation strategy: *Soft-Rejection Sampling* (GIFT-REJECT), which adds diverse geometrically valid alternative programs as new output targets, and Failure-Driven Augmentation (GIFT-FAIL), which renders near-miss predictions into synthetic inputs paired with the original ground-truth code.

- We show that this augmentation pipeline expands the effective training set substantially, improves single-shot accuracy over a strong supervised baseline, and reduces the reliance on expensive test-time sampling.

- We demonstrate through extensive analysis that GIFT improves robustness on complex geometries, narrows the amortization gap between pass@1 and pass@k, and remains competitive with more complex multimodal systems without requiring additional human annotation or specialized architectures.

## 2. Background

**Vision-Language Models** Vision-Language Models (VLMs (Liu et al., 2024; Bai et al., 2025)) are powerful foundation models that integrate Large Language Models (LLMs) with vision encoders to process visual information (Fig. 4). These systems are highly versatile, supporting diverse tasks such as visual captioning, QA, multitasking, referring expressions, and code generation. Although aligning visual and textual components remains an open challenge, recent VLMs based on Qwen have established strong baselines.

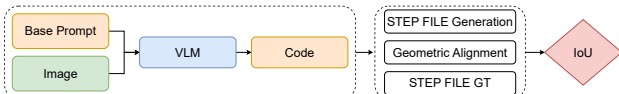

*Figure 3.* The CAD-Coder pipeline processes multimodal inputs (text prompts and images) to generate executable CAD code. This code is converted into STEP files via geometric alignment and validated against the ground truth using IoU metrics.

**Image-Conditional CAD Program Synthesis** Image-to-CAD (Alam &Ahmed, 2024; Li et al., 2025c; You et al., 2025) remains a central challenge in generative design. Recently, CAD-Coder (Doris et al., 2025) introduced an end-to-end pipeline to bridge the gap between 2D images and parametric code (Fig. 3). Its three-stage workflow utilizes a visual encoder (e.g., LLaVA (Liu et al., 2024) or Qwen-VL (Bai et al., 2025)) to project geometric features into

language embeddings, employs an autoregressive LLM to generate executable CadQuery scripts, and executes the code via a Python interpreter to reconstruct B-Rep models. The system is trained via Supervised Fine-Tuning (SFT) on the 160k image-code pairs of the GenCAD-Code dataset (Alam &Ahmed, 2024; Doris et al., 2025). The training optimizes a standard next-token prediction objective:

$$\mathcal{F}_{SFT}(\theta) = \mathbb{E}_{(\mathbf{x},\mathbf{c})\sim\mathcal{D}_{SFT}}\left[\log p_\theta(\mathbf{x}|\mathbf{c})\right]$$
$$= \sum_{\mathbf{c}}\sum_{t=1}^{T}\log p_\theta(\mathbf{x}_t \mid \mathbf{x}_{<t}, \mathbf{c}), \quad (1)$$

which aligns visual semantics with strict `CadQuery` syntax using a standard MLE estimator. Here, **c** represents the input image and a fixed text prompt (`"generate cadquery code for the image"`). And **x** represents the model output, i.e. code or a CAD format. We adopt this baseline algorithm to train our GIFT models across varying data sources.

**Limitations** Current methods face significant hurdles. First, SFT is brittle: it enforces a rigid one-to-one mapping that penalizes valid alternative programs yielding the same geometry, while also relying on scarce, low-diversity datasets. Second, RL is complex and fragile; it requires intricate training pipelines and specialized infrastructure, making reproduction difficult. Finally, scaling is computationally inefficient due to hardware bottlenecks; specifically, reward calculation relies on CPU-based CAD solvers, the constant communication overhead with GPU-based models severely limits performance scaling. See Appendix A for additional discussion.

## 3. Method

We argue that the main bottleneck in image-conditional CAD generation is not model capacity, but *modality alignment*: the scarcity of diverse supervised examples that connect visual geometry to valid program structure. To address this limitation, we introduce GIFT, a verifier-guided data augmentation framework. Rather than relying solely on the original image-code pairs, GIFT samples candidate programs from a pretrained image-to-CAD model, evaluates them with a geometric kernel, and converts geometrically valid and near-valid generations into additional supervised training examples.

Operationally, GIFT transforms inference-time search into an offline augmentation pipeline (Algorithm 1). For each training image, high-quality alternative programs are retained as additional targets for the original input, while structured near-miss failures are rendered back into the image domain and paired with the ground-truth code. The resulting augmented dataset improves both output diver-

sity and robustness without requiring online reinforcement learning or additional human annotation.

**Program Synthesis as Latent Reasoning** While multiple distinct programs can generate the same correct output, most engineering datasets provide only a one-to-one mapping. Consequently, it is challenging to steer the generative process to encourage diversity while simultaneously respecting the underlying data distribution. This creates an unnecessarily narrow training signal.

We model Image-to-CAD generation as a conditional latent-program problem, where the goal is to learn a distribution over programs $\mathbf{z}$ that can decode to the correct geometry $\mathbf{x}$ rather than a single canonical string:

$$p_\theta(\mathbf{x}, \mathbf{z}|\mathbf{c}) = p(\mathbf{x}|\mathbf{z}, \mathbf{c})\, p_\theta(\mathbf{z}|\mathbf{c}), \tag{2}$$

where $\mathbf{x}$ denotes the decoded CAD representation (B-rep, mesh), $\mathbf{z}$ represents the latent program (in `cadquery` syntax), and $\mathbf{c}$ is the conditioning input, consisting of a single image and a short, fixed textual prompt. For CAD reconstruction, the conditional distribution $p(\mathbf{x}|\mathbf{z}, \mathbf{c})$ can be approximated as a deterministic decoding step $\mathbf{x} = d(\mathbf{z}, \mathbf{c})$. This decoding is typically non-learned, relying on standard CAD tools to convert code into formats such as STEP files. Therefore, image-conditional program synthesis focuses on learning an expressive $p_\theta(\mathbf{z}|\mathbf{c})$, typically via SFT and RL.

**Inference-Time Scaling for CAD Generation** We use Inference-Time Scaling (ITS (Brown et al., 2024; Snell et al., 2024)) primarily as a data-generation tool rather than as a deployment strategy. For each training image, we sample multiple candidate programs from the base model and verify them against the ground-truth geometry using a CAD kernel. The verified samples are then partitioned into three groups: exact or near-exact solutions, recoverable near-miss failures, and unrecoverable failures. This post-hoc analysis enables us to construct an augmented training set that both broadens the target distribution and explicitly trains on hard cases.

This procedure provides a form of weak supervision. Instead of treating only exact string matches as correct, we retain any program that produces sufficiently accurate geometry. In doing so, we shift training toward better coverage of the valid solution space rather than a single canonical program:

$$\{\mathbf{z}^{(k)}\}_{k=1}^K \sim p_\theta(\mathbf{z}|\mathbf{c}). \tag{3}$$

Crucially, because this feedback requires a CAD kernel and ground-truth programs for verification, we restrict augmentation to the original training distribution and do not introduce any additional human-crafted supervision.

**Bootstrapping** We generate candidate programs using `QwenVL-2.5-7B-CadCoder` (Doris et al., 2025), utilizing ground truth STEP files in the GenCad-Code trainset

for geometric feedback. To ensure sampling diversity, we employ a scaling strategy across five computational budgets ($N \in \{8, 16, 32, 64, 128\}$). We apply an inverse hyper-parameter strategy: lower budgets use wider temperature ranges to maximize exploration, while higher budgets enforce strict low-temperature sampling to prioritize precision. Full sampling configurations are detailed in Table 11, Appendix F and I.

### 3.1. Geometric Inference Feedback Tuning

Let $\mathcal{D}_{SFT} = \{(\mathbf{c}^{(n)}, \mathbf{z}_{gt}^{(n)})\}_{n=1}^N$ be a training dataset of $N$ image-program pairs, where $\mathbf{c}$ is the visual input and $\mathbf{z}_{gt}$ is the ground truth CAD program (Fig. 5). We denote the base policy (the base Image-to-Code model (Doris et al., 2025)) as $p_\theta(\mathbf{z}|\mathbf{c})$.

For each input $\mathbf{c}$, we perform inference-time scaling by sampling a set of $K$ candidate programs $\mathcal{Z}_K$ from the current policy:

$$\mathcal{Z}_K = \{\mathbf{z}^{(k)}\}_{k=1}^K \sim p_\theta(\mathbf{z}|\mathbf{c}).$$

For every candidate $\mathbf{z} \in \mathcal{Z}_K$, we compute a geometric validity function $f(\mathbf{z})$ using a CAD kernel (e.g., OpenCAS-CADE) to measure alignment with the ground truth:

$$f(\mathbf{z}) = \text{IoU}(\text{Execute}(d(\mathbf{z}))), \text{Execute}(d(\mathbf{z}_{gt}))),$$

where $d(\mathbf{z})$ projects code to a format amenable to the CAD kernel (STEP or STL files), and Execute runs the geometric kernel to obtain the final 3D model as a B-Rep or Mesh representation, and the Intersection-over-Union (IoU) is computed between the generated and ground truth CAD models. In the following we will use the shorthand $f(\mathbf{z}) :=$ $\text{IoU}(\mathbf{z}, \mathbf{z}_{gt})$ for simplicity.

We employ two complementary augmentation mechanisms: Soft Rejection Sampling (SRS) and Failure-Driven Augmentation (FDA). SRS augments the output space (code) by capturing diverse valid programs to limit memorization, while FDA augments the input space (image) to specifically target hard failures.

Through repeated sampling, this approach constructs an augmented training set whose composition reflects the model's inference behavior across compute budgets.

**Threshold Selection** We choose the filtering thresholds empirically from the cumulative distribution of inference-time IoU scores (Fig. 9 and 8). We set $\tau_{low} = 0.5$ because roughly 10% of generated programs fall below this level; these samples are typically degenerate or non-executable and are excluded from the training pool. We set $\tau_{valid} = 0.9$ to separate high-quality solutions from recoverable failures. Approximately 40% of samples fall below this threshold.

This yields a natural dual strategy: Soft-Rejection Sampling

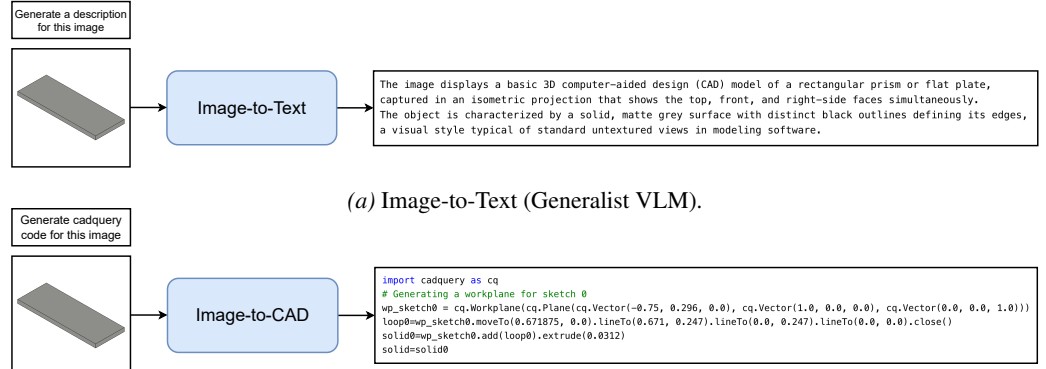

*(a)* Image-to-Text (Generalist VLM).

*(b)* Image-to-CAD (Specialist VLM).

*Figure 4.* Comparison of Vision-Language Model generations: (a) standard text description vs. (b) executable CAD code.

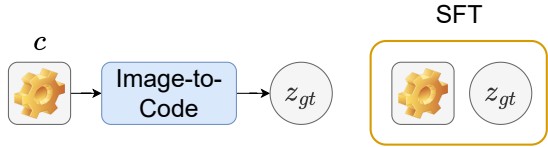

*Figure 5.* Standard SFT Baseline. The model is trained on static image-code pairs using a next-token prediction objective.

uses the high-fidelity tail ($\text{IoU} \geq 0.9$), while Failure-Driven Augmentation targets near-miss failures ($0.5 \leq \text{IoU} < 0.9$).

**Soft Rejection Sampling: Output Augmentation** Standard rejection sampling typically selects only the single best sample or strictly correct programs ($f(\mathbf{z}) = 1$). This limits diversity. To overcome this, SRS utilizes the feedback from the set $\mathcal{Z}_K$ to identify a broader set of valid programs (Fig. 6). We define a selection indicator $w_{srs}(\mathbf{z})$ that filters

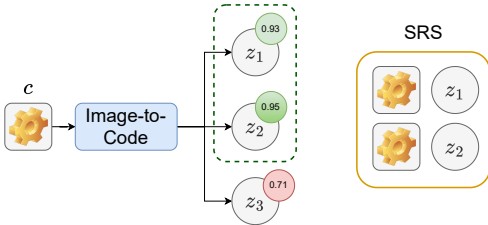

*Figure 6.* Soft Rejection Sampling (SRS). We sample $K$ programs, verify them with a geometric kernel, and retain diverse valid programs ($0.9 \leq \text{IoU} < 0.99$) to expand the training distribution beyond the single ground truth.

for high-fidelity alternatives that are distinct from the exact ground truth:

$$w_{srs}(\mathbf{z}) = \mathbb{1}\left[\tau_{valid} \leq f(\mathbf{z}) < \tau_{match}\right],$$

where $\tau_{valid} = 0.9$ and $\tau_{match} = 0.99$. The resulting SRS dataset $\mathcal{D}_{SRS}$ expands the training targets by including these

diverse valid solutions:

$$\mathcal{D}_{SRS} = \bigcup_{(\mathbf{c},\mathbf{z})\in\mathcal{D}} \left\{(c, \mathbf{z}) \mid \mathbf{z} \in \mathcal{Z}_K,\ w_{srs}(\mathbf{z}) = 1\right\}.$$

This procedure broadens the training distribution by including multiple high-quality programs for the same input. As a result, the model is less likely to collapse onto a single rigid syntactic pattern.

This trades exact string matching for greater diversity and coverage in program space. The threshold $\tau_{valid}$ preserves geometric quality while still allowing meaningful variation in program form. We follow the analysis in Fig. 9b to choose the thresholds for $\tau$.

**Exploiting Geometric Feedback** Although soft labeling and output augmentation are well established in self-training and weak-supervision settings (Dong et al., 2023; Zelikman et al., 2024), GIFT differs in how it obtains the signal. Instead of relying on heuristic confidence scores or learned rewards, it uses a deterministic geometric kernel to identify novel, valid alternative programs. This makes it possible to expand the target distribution with high-quality synthetic supervision drawn directly from the original training set.

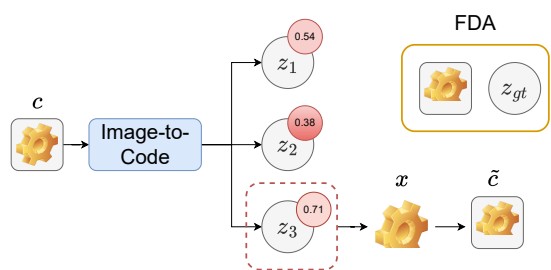

*Figure 7.* Failure-Driven Augmentation (FDA). We identify hard negatives (programs with $0.5 \leq \text{IoU} < 0.9$), render them into synthetic images ($\phi(\mathbf{z})$), and train the model to map these 'noisy' inputs back to the correct ground truth code ($\mathbf{z}_{gt}$).

**Learning from Failure: Input Augmentation** While SRS reinforces successful and diverse outputs, it does not directly address persistent failure cases. To do so, we use programs that are geometrically plausible yet remain incorrect across both small and large sampling budgets in $\mathcal{Z}_K$. These near-miss samples provide a useful signal because they encode structured errors rather than random noise. We define a failure indicator $w_{fda}(\mathbf{z})$ to select these hard negatives:

$$w_{fda}(\mathbf{z}) = \mathbb{1}\left[\tau_{low} \le f(\mathbf{z}) < \tau_{valid}\right],$$

where $\tau_{low} = 0.5$ (see analysis in Fig. 9b). Unlike SRS, which pairs the sampled code with the original image, FDA uses the sampled code to construct a challenging input (Fig. 7). We define a *rendering function* $\phi$ that projects the CAD model back into the input image domain, i.e. $\tilde{\mathbf{c}} \leftarrow \phi(d(\mathbf{z}))$.

Since $\mathbf{z}$ is a valid executable program, we can render it into a synthetic image $\tilde{\mathbf{c}}$. This synthetic image represents the visual manifestation of the model's geometric error. The FDA dataset $\mathcal{D}_{FDA}$ consists of pairs mapping the failed visual state back to the original ground truth:

$$\mathcal{D}_{FDA} = \bigcup_{(\tilde{\mathbf{c}}, \mathbf{z}_{gt}) \in \mathcal{D}} \{(\phi(d(\mathbf{z})), \mathbf{z}_{gt}) \mid \mathbf{z} \in \mathcal{Z}_K,\ w_{fda}(\mathbf{z}) = 1\}.$$

This turns FDA into a geometric denoising objective: the model receives an imperfect rendered input $\tilde{\mathbf{c}}$ and is trained to recover the corresponding ground-truth program $z_{gt}$. In practice, this improves robustness to visual ambiguity and partial geometric mismatch.

**Objective** The final GIFT objective amortizes the expensive inference-time search into the model weights by optimizing the log-likelihood over the combined augmented datasets:

$$\mathcal{F}_{\text{GIFT}}(\theta) = \underbrace{\mathbb{E}_{(\mathbf{c}, \mathbf{z}) \sim \mathcal{D}_{SFT}}[\log p_\theta(\mathbf{z}|\mathbf{c})]}_{\text{Base (SFT)}} + \underbrace{\mathbb{E}_{(\mathbf{c}, \mathbf{z}) \sim \mathcal{D}_{SRS}}[\log p_\theta(\mathbf{z}|\mathbf{c})]}_{\text{Output Diversity (SRS)}} + \underbrace{\mathbb{E}_{(\mathbf{c}, \mathbf{z}) \sim \mathcal{D}_{FDA}}[\log p_\theta(\mathbf{z}|\mathbf{c})]}_{\text{Input Robustness (FDA)}}.$$

(4)

Note that $w_{srs}$ and $w_{fda}$ operate on *disjoint intervals* of IoU (valid vs. near-miss), ensuring distinct gradient signals. In practice, sampling images $\mathbf{c}^{(n)}$ and programs $\mathbf{z}_{gt}^{(n)}$ from the training set $\mathcal{D}_{SFT}$, and generating $K$ Monte Carlo samples from the base model for each $\mathbf{c}^{(n)}$, $\{\mathbf{z}^{(k)}\}_{k=1}^K \sim p_\theta(\mathbf{z}|\mathbf{c}^{(n)})$,

we obtain the empirical objective:

$$\tilde{\mathcal{F}}_{\text{GIFT}}(\theta) \propto \sum_{n=1}^N \log p_\theta(\mathbf{z}_{gt}^{(n)}|\mathbf{c}^{(n)}) +$$
$$\sum_{n=1}^N \sum_{k=1}^K w_{srs}(\mathbf{z}^{(k)}) \log p_\theta(\mathbf{z}^{(k)}|\mathbf{c}^{(n)}) + \qquad (5)$$
$$\sum_{n=1}^N \sum_{k=1}^K w_{fda}(\mathbf{z}^{(k)}) \log p_\theta(\mathbf{z}_{gt}^{(n)}|\tilde{\mathbf{c}}^{(k)}),$$

where we omit the index $n$ over $\mathbf{z}^{(k)}$ for clarity. $\tilde{c}^{(k)}$ here is the output of the inverse mapping $\phi(d(\mathbf{z}^{(k)}))$ and $\mathbf{z}^{(k)} \sim p_\theta(\mathbf{z}|\mathbf{c}^{(n)})$ is a sample from the model conditioning on an image. The final training loss is minimized as $\mathcal{L}(\theta) = -\tilde{\mathcal{F}}_{\text{GIFT}}(\theta)/N$. We provide a detailed gradient analysis of this objective in Appendix G.

## 4. Experiments

**Dataset** We conduct our evaluation on the GenCAD-Code dataset (Alam & Ahmed, 2024), derived from the DeepCAD dataset (Wu et al., 2021), which contains 163k image-code pairs of parametric CAD designs for train, and 8k image-code pairs for testing. We additionally evaluate on a curated set of 400 out-of-distribution (OOD) samples (Doris et al., 2025) that present realistic input images not well-represented in the training distribution.

**Baselines** We use three primary vision-language models: Qwen-VL-2.5-7B-Instruct and Qwen-VL-2.5-3B-Instruct as base multimodal foundation models, and CAD-Coder-SFT (7B) as the main supervised fine-tuning baseline and sampling model. The CAD-Coder-SFT model serves as our starting point for exploring geometric inference feedback tuning. In the following we refer to CAD-Coder-SFT as SFT.

**Setup** We evaluate three training configurations: (i) standard Supervised Fine-Tuning (SFT) as our baseline, (ii) GIFT with Soft-Rejection Sampling (SRS), which captures diverse valid programs beyond ground-truth matching, and (iii) GIFT with both SRS and Failure-Driven Augmentation (FDA), which additionally targets hard examples where the base model consistently struggles. Through our SRS and FDA strategies, we expand the original 163k training set to approximately 370k samples (Appendix F and I). We train all models for a maximum of 10 epochs or when performance stops improving for two consecutive epochs. Empirically, models converged within 3 to 4 epochs.

In the following, we use GIFT-REJECT to denote the model trained with SRS, GIFT-FAIL for the one trained with FDA, and GIFT to denote the model trained with both SRS and FDA.

**Evaluation and Geometric Verifier** We use IoU as main metric as proposed in (Doris et al., 2025). GIFT relies heavily on geometric kernels (OpenCASCADE) for IoU computation. For the specific calculation, we follow the IoU-best protocol from (Doris et al., 2025; Li et al., 2025c). In particular, we first execute the generated CadQuery script to obtain a Boundary Representation (B-Rep) solid. We then normalize both the generated and ground-truth solids by centering and scaling them based on the trace of their inertia tensor. To address the ambiguity in orientation, we align their principal axes and compute the Intersection-over-Union (IoU) across different axis permutations, defining IoU-best as the maximum value obtained. This ensures the verification is robust to rigid transformations and coordinate system mismatches.

## 4.1. Inference-Time Scaling: Analysis

We first investigate the latent capabilities of the base CAD-Coder model by analyzing its scaling behavior on the training set. We employ Inference-Time Scaling (ITS) with varying computational budgets $N \in \{8, 16, 32, 64, 128\}$, generating multiple program candidates per image and filtering them via geometric verification (IoU against ground truth).

*Table 1.* Performance comparison between Top-10% and Top-1 inference-time selections on the GenCAD training subset. We report the mean IoU obtained by retaining either the top 10% of generated samples or only the single best sample per input, with and without exact ground-truth matches.

| Exp | Condition | Budget $N$ | Input (Image) | Output (Code) | Mean |
|---|---|---|---|---|---|
| IoU Top 10% | w/ IoU=1 | (8, 16, 32, 64, 128) | 4k | 169,116 | 0.734 |
| | w/o IoU=1 | (8, 16, 32, 64, 128) | 4k | 134,345 | 0.665 |
| IoU Best (Top1) | w/ IoU=1 | (8, 16, 32, 64, 128) | 4k | 3,911 | 0.839 |
| | w/o IoU=1 | (8, 16, 32, 64, 128) | 4k | 2,619 | 0.760 |

**Performance Gap** As shown in Table 1, there is a significant disparity between the model's average and peak performance. While the mean IoU for the top 10% of samples is 0.734, selecting the single best sample (Top-1) boosts performance to 0.839. This gap indicates that the model frequently generates valid solutions that are discarded during standard greedy decoding, confirming that significant learning signal remains unexploited within the model's weights.

**Failure Modes** Figure 8 visualizes the IoU distribution of generated samples. Even when filtering for the best candidates (Fig. 8c), the distribution remains bimodal: a large cluster of high-fidelity solutions contrasts with a persistent tail of failures where the model cannot recover correct geometry regardless of sampling budget. Notably, removing exact ground-truth matches (Fig. 8d) reveals a rich set of alternative valid programs ($0.9 < \text{IoU} < 0.99$), which we

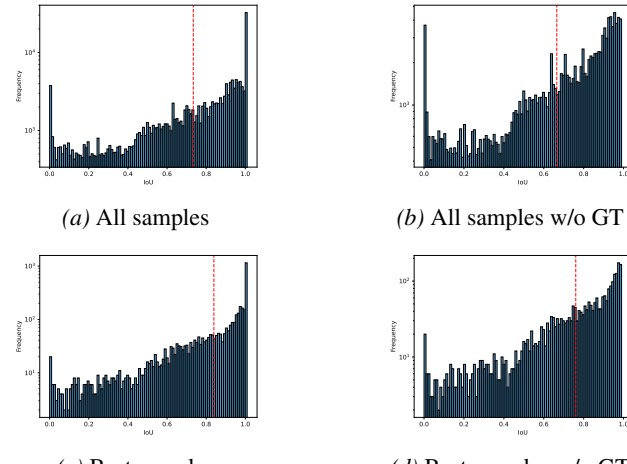

*(a)* All samples      *(b)* All samples w/o GT

*(c)* Best samples      *(d)* Best samples w/o GT

*Figure 8.* IoU distributions for samples obtained via ITS, selecting the top 10% and top-1 programs per image. Panels show: (a) all samples; (b) all samples excluding ground truth (GT) matches; (c) best-performing samples; and (d) best samples excluding GT. The persistence of low-quality generations, even in the best-performing subset, highlights the need to amortize offline ITS feedback to improve base model reliability.

can leverage to improve data coverage.

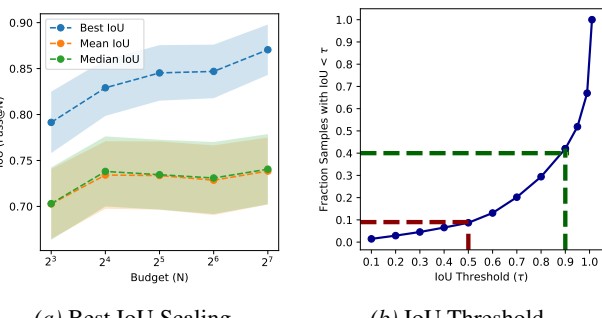

*(a)* Best IoU Scaling.      *(b)* IoU Threshold.

*Figure 9.* Analysis of inference-time IoU statistics. (a) Pass@N scaling shows that best-case performance improves with compute budget, while mean and median gains saturate more quickly. (b) The cumulative IoU distribution motivates the filtering thresholds used for SRS and FDA by separating low-quality failures from high-fidelity solutions

**Scaling Limits** Figure 9 demonstrates that while performance improves with budget $N$, returns diminish rapidly. Simple stochastic sampling often introduces syntactic errors rather than semantic diversity. This suggests that merely scaling inference is inefficient; instead, we must amortize these expensive successful search paths back into the model.

## 4.2. Amortizing Inference-Time Augmentation

The significant performance disparity between Top-1 and Top-10% generations drives our inference-time compute-driven data augmentation strategy. To tune the model, we leverage high-fidelity candidates via *Soft-Rejection Sam-*

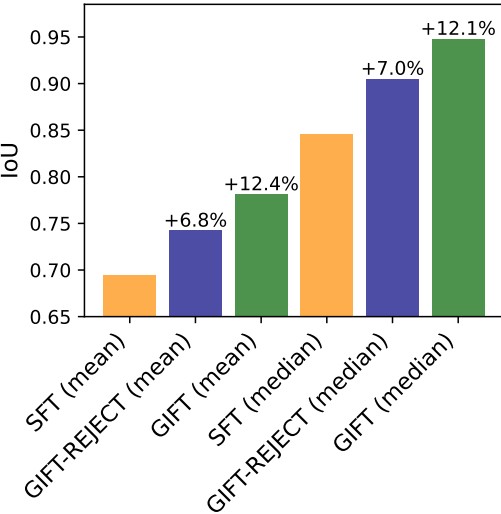

*Figure 10.* GIFT strategies impact. We compare the SFT baseline against GIFT-REJECT (targeting output diversity) and the full GIFT framework (combining diversity and robustness). The full GIFT method yields the largest performance gain, confirming the critical value of learning from hard negatives.

*pling* to expand solution coverage, while utilizing *Failure-Driven Augmentation* to specifically target inputs where the model consistently fails.

**Efficiency** We further analyze the computational efficiency of our method by comparing the inference-time scaling curves (Figure 1). The GIFT model consistently outperforms the SFT baseline at every level of compute budget (Table 2). GIFT matches the peak performance of the SFT model (achieved via extensive rejection sampling) while reducing the inference compute requirement by approximately 80% (Figure 1). This confirms that GIFT successfully distills the benefits of test-time search directly into the model weights. An extended version of this scaling analysis, including the GIFT-FAIL variant, is provided in Appendix C (Figure 12), where the same ordering holds across inference budgets.

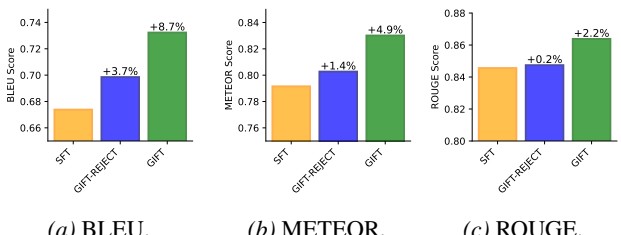

*(a)* BLEU.   *(b)* METEOR.   *(c)* ROUGE.

*Figure 11.* Evaluation results on standard captioning metrics for SFT and GIFT variants.

**Performance** We evaluate the impact of these strategies on the test set, as shown in Figure 10. The baseline SFT

model achieves a mean IoU of 0.698. The GIFT-REJECT strategy significantly boosts this performance by exposing the model to a diverse set of valid programs. Adding Failure-Driven Augmentation (full GIFT augmentation framework) further improves generalization, increasing mean IoU from 0.698 to 0.782.

We also observe consistent gains on standard text-generation metrics, including BLEU, METEOR, and ROUGE (Figure 11). These improvements suggest that GIFT better aligns generated program syntax with the distribution of valid target code.

**Amortization Gap** We quantify the amortization gap as the difference between single-shot performance (pass@1) and oracle performance under repeated sampling (pass@k). As shown in Table 3, the SFT baseline has a gap of 15.5%, indicating substantial reliance on rejection sampling to achieve high performance. GIFT-REJECT reduces this gap to 10.9%, while the full GIFT model reduces it further to 5.2%, a 66.4% relative reduction compared with the baseline.

These results show that GIFT shifts probability mass toward high-quality outputs, improving performance even without extensive test-time sampling.

### 4.3. Intrinsic Metrics

To further analyze the robustness of our approach, we examine the relationship between model performance and task complexity, where complexity is proxied by the token length of the ground truth CAD programs (Fig. 2b). As illustrated in Fig. 2c, while all models exhibit an expected degradation in performance as the geometric complexity increases, the models trained with GIFT demonstrate significantly greater resilience compared to the SFT baseline.

The performance gap becomes particularly pronounced at higher complexity levels, where the baseline SFT model's accuracy drops sharply. In contrast, GIFT maintains a more stable performance profile, effectively mitigating the curse of complexity often observed in autoregressive code generation. This suggests that by incorporating hard negatives through Failure-Driven Augmentation (FDA) and diverse valid solutions via Soft-Rejection Sampling (SRS), the model learns to sustain geometric coherence even for intricate designs requiring longer program sequences. Appendix D expands this analysis with finer-grained complexity breakdowns and distributional statistics over operations and token lengths (Figure 17 and 18).

### 4.4. General Evaluation

Table 4 compares GIFT with recent modality-to-CAD systems across several input settings, including depth, point

*Table 2.* Test-set IoU across increasing inference-time compute budgets for CAD-Coder-SFT, GIFT-REJECT, GIFT-FAIL, and the full GIFT model. The final row reports the relative improvement of GIFT over the SFT baseline at each budget.

| Method | 1 | 2 | 3 | 4 | 5 | 6 | 7 | 8 | 9 | 10 |
|---|---|---|---|---|---|---|---|---|---|---|
| SFT | 0.698 | 0.725 | 0.743 | 0.763 | 0.780 | 0.788 | 0.792 | 0.801 | 0.804 | 0.807 |
| GIFT-REJECT | 0.732 | 0.745 | 0.762 | 0.773 | 0.785 | 0.796 | 0.802 | 0.807 | 0.810 | 0.812 |
| GIFT-FAIL | 0.761 | 0.780 | 0.789 | 0.791 | 0.799 | 0.802 | 0.802 | 0.803 | 0.803 | 0.806 |
| GIFT | 0.779 | 0.802 | 0.809 | 0.811 | 0.812 | 0.816 | 0.816 | 0.817 | 0.818 | 0.819 |
| $\Delta(GIFT, SFT)$ | +11.60 % | +10.53 % | +8.92 % | +6.31 % | +4.04 % | +3.52 % | +3.02 % | +2.01 % | +1.80 % | +1.56 % |

*Table 3.* Mean IoU on the GenCAD test set and amortization gap between pass@1 and pass@k under different dataset-augmentation strategies. Smaller gaps indicate that model quality is better amortized into single-shot generation.

| Method | pass@1 | pass@5 | pass@10 | Gap | $\Delta$ |
|---|---|---|---|---|---|
| SFT | 0.698 | 0.776 | 0.807 | 15.5 % | - |
| GIFT-REJECT | 0.732 | 0.788 | 0.812 | 10.9 % | -29.6 % |
| GIFT-FAIL | 0.761 | 0.792 | 0.806 | 5.9 % | -61.9 % |
| GIFT | 0.777 | 0.812 | 0.820 | 5.2 % | -66.4 % |

clouds, text, multi-view images, and single-view images. Despite using only a single-view image as input, GIFT delivers competitive performance relative to methods that rely on richer modalities such as point clouds or multi-view supervision. In Table 9 we provide evidence of sensitivity to failed rendering (code that cannot be compiled) for modality-to-code models, and show how median values are more reliable and stable than mean values for such problems.

*Table 4.* Comparison with recent modality-to-CAD systems trained on DeepCAD variant datasets under single-sample inference. Inputs include point clouds (PC), text (TXT), multi-view images (IMG), and single-view images (IMG-s).

| | Modality | IoU |
|---|---|---|
| *frontier zero-shot* | | |
| GPT4o | IMG-s | 0.378 |
| Gemini-2.5-PRO | IMG-s | 0.451 |
| Gemini-3.0-PRO | IMG-s | 0.540 |
| *open-source zero-shot* | | |
| Qwen-2.5-72B (Bai et al., 2025) | IMG-s | 0.345 |
| *open-source finetuned* | | |
| PointNet (Wu et al., 2021) | PC | 0.467 |
| PointBERT (Xu et al., 2023) | PC | 0.653 |
| TransCAD (Dupont et al., 2024) | PC | 0.655 |
| PrismCAD (Lambourne et al., 2022) | PC | 0.721 |
| Point2Cyl (Uy et al., 2022) | PC | 0.738 |
| CAD-Diffuser (Ma et al., 2024) | PC | 0.743 |
| CAD-SigNet (Khan et al., 2024a) | PC | 0.773 |
| Text2CAD (Khan et al., 2024b) | TXT | 0.715 |
| Cadrille (Kolodiazhnyi et al., 2025) | PC | 0.794 |
| Cadrille-RL (Kolodiazhnyi et al., 2025) | PC,IMG,TXT | 0.859 |
| GACO-CAD-RL (Wang et al., 2025b) | IMG-s,Depth,Normal | 0.864 |
| ReCAD-VL (Li et al., 2025c) | IMG-s | 0.631 |
| CAD-Coder-SFT (Doris et al., 2025) | IMG-s | 0.691 |
| CAD-Coder-GIFT (ours) | IMG-s | 0.782 |

**Ablations** Table 5 isolates the contribution of each component of GIFT. Naive augmentation provides only marginal gains, whereas GIFT-REJECT improves performance by mining diverse valid programs and GIFT-FAIL improves robustness by targeting recoverable errors in the 0.5-0.9 IoU range. Combining both mechanisms yields the strongest overall performance, confirming that diversity and failure correction contribute complementary benefits.

Extended ablations in Appendix C confirm these benefits persist across diverse model architectures (Fig. 13) and sampling temperatures (Fig. 14), with strong generalization to out-of-distribution geometries.

Finally, in Fig. 15 we run preliminary experiments to evaluate out-of-distribution performance and assess generalization beyond the in-distribution setting using real world images as input. Table 7 provides valid samples rates and error analysis for in- and out-of-distribution test splits.

*Table 5.* Ablation of GIFT variants on the GenCAD test set. AUG denotes standard augmentation.

| Method | IoU |
|---|---|
| SFT (LONG) | 0.698 |
| SFT w/ AUG | 0.710 |
| GIFT-REJECT | 0.742 |
| GIFT-REJECT w/ AUG | 0.745 |
| GIFT-FAIL | 0.761 |
| GIFT | 0.782 |

## 5. Limitations and Conclusion

We presented *Geometric Inference Feedback Tuning* (GIFT), a verifier-guided data augmentation framework for Image-to-CAD program synthesis. By using a CAD kernel to mine both diverse valid outputs and structured failure cases, GIFT converts inference-time search into additional supervised training signal. Empirically, this produces stronger single-shot generation, reduces dependence on inference compute, and narrows the gap to more complex multimodal systems.

**Limitations** Our approach relies on a deterministic geometric kernel and ground-truth CAD programs to verify generated samples, which currently limits its application in purely unsupervised or "in-the-wild" settings where such verifiers are unavailable. While GIFT significantly reduces inference latency, the bootstrapping phase incurs a one-time offline computational cost to generate and filter the candidate programs.

## Impact Statement

This paper presents work whose goal is to advance the field of machine learning, specifically within the domain of neuro-symbolic program synthesis and automated Computer-Aided Design (CAD). Broadly, this work aims to democratize engineering workflows by enabling the rapid conversion of 2D images into editable parametric 3D models. While this lowers the barrier to entry for digital manufacturing and design, we acknowledge the potential for such technologies to impact labor markets in drafting-heavy roles or to be misused for intellectual property infringement through automated reverse engineering. However, these challenges are inherent to the broader field of generative design, and we do not believe our specific method introduces new ethical risks that require unique mitigation strategies beyond standard responsible AI practices.

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

# A. Related Work

**Generative Models for CAD** Generative design is experiencing a widespread resurgence across engineering domains, from topology optimization to mechanism synthesis (Regenwetter et al., 2022; Mazé &Ahmed, 2023; Nobari et al., 2024). Early approaches to CAD generation focused on direct 3D synthesis, producing outputs such as point clouds, voxels, boundary representations, and meshes (Lambourne et al., 2021; Li et al., 2025b; Alam &Ahmed, 2024; Xu et al.; 2025; Jayaraman et al., 2022). However, these topological formats are typically too complex for engineers to easily parameterize or edit (Yu et al., 2025). To address this limitation, recent research has shifted toward *program synthesis* (Bhuyan et al., 2024; Jones et al., 2022; Ganeshan et al., 2023), where models output executable code (e.g., CadQuery) to construct the desired geometry (Doris et al., 2025; Kolodiazhnyi et al., 2025; Rukhovich et al., 2025). While promising, these methods often rely on post-training pipelines hindered by scarce multimodal data. This scarcity leads to brittle alignment between inputs and the resulting programs, ultimately requiring heavy manual engineering and post-processing (Guan et al., 2025; Wang et al., 2025b; Chen et al., 2025; Li et al., 2025d; Niu et al., 2026). Although some frameworks attempt to resolve these alignment issues using online reinforcement learning, they introduce computational overhead; executing CPU-bound geometric kernels within the training loop creates data-loading bottlenecks that starve GPU resources (Li et al., 2025c; Kolodiazhnyi et al., 2025). GIFT overcomes this bottleneck by decoupling exploration from training, running the geometric kernel offline, and heavily parallelizing the sampling process. Furthermore, while other recent studies use feedback-driven techniques to improve post-training (Wang et al., 2025a; Ding et al., 2026), GIFT integrates this feedback directly into inference-time compute to dynamically steer the sampling process.

**Inference-Time Scaling** Inference-Time Scaling (ITS) enhances generative capabilities by allocating additional compute at test time to produce higher-quality outputs, effectively framing generation as a guided search problem (Wang et al., 2022; Wei et al., 2022; Brown et al., 2024; Snell et al., 2024; Beeching et al., 2024; Kang et al., 2025; Dang et al., 2025). Within this framework, two primary strategies have emerged: sequential refinement techniques that iteratively improve a single output (Shinn et al., 2024; Yao et al., 2022; Jaech et al., 2024; Guo et al., 2025), and parallel approaches - such as Self-Consistency and Best-of-N sampling - that generate multiple candidates simultaneously (Wang et al., 2022; Chen et al., 2023; Brown et al., 2024; Kang et al., 2025; Amini et al., 2024). To steer this search toward optimal solutions, both methods typically rely on outcome verifiers or process rewards (Lightman et al., 2023; Cobbe et al., 2021; Puri et al., 2025; Geuter et al., 2025; Giannone et al., 2025). Particularly in code generation, execution feedback serves as a critical mechanism to scale inference (Li et al., 2025a) and boost overall performance (Li et al., 2022). GIFT builds upon these foundational concepts but focuses on *amortizing* the search process. By distilling the successes of expensive test-time computation directly into the model's weights, GIFT achieves efficient, high-quality generation without the burden of added inference latency.

**Self-Training and Data Augmentation** Iterative self-training methods - such as STaR (Zelikman et al., 2022), ReST (Gulcehre et al., 2023), and RFT (Xiong et al., 2025) - have demonstrated that models can bootstrap their own performance by training on filtered, high-quality generations. Whereas reasoning domains typically rely on logical consistency checks for this filtering, CAD generation is uniquely guided by a deterministic geometric kernel rather than sparse unit tests or learned reward models. We leverage this kernel as a dense verifier to provide exact, sample-level feedback. Furthermore, while frameworks like RFT (Xiong et al., 2025) boost performance by training exclusively on "gold" samples, they inadvertently discard the valuable learning signals embedded in near-miss failures. GIFT generalizes this approach by not only reinforcing diverse successes (via SRS) but also explicitly learning to correct failures (via FDA). By pairing rendered failure cases with their corresponding ground-truth code, we effectively transform geometric errors into a supervised signal for geometric denoising (Bsharat &Shen, 2025). Unlike standard online reinforcement learning techniques that demand complex value estimation and online sampling (Schulman et al., 2017; Guo et al., 2025), GIFT amortizes the search cost directly into the supervised policy. This approach elevates the training dataset from a static collection of successes into a dynamic curriculum of error correction.

# B. Algorithm

---

**Algorithm 1** Geometric Inference Feedback Tuning (GIFT)

---

**Input:** Dataset $\mathcal{D}_{SFT} = \{(\mathbf{c}^{(n)}, \mathbf{z}_{gt}^{(n)})\}_{n=1}^{N}$, Base Model $p_\theta$, Budget $K$
**Parameters:** $\tau_{low} = 0.5$, $\tau_{valid} = 0.9$, $\tau_{match} = 0.99$
**Initialize:** $\mathcal{D}_{train} \leftarrow \mathcal{D}_{SFT}$

// Data: Bootstrapping via Inference-Time Scaling
**for** each $(\mathbf{c}, \mathbf{z}_{gt}) \in \mathcal{D}_{SFT}$ **do**
    Sample candidates $\mathcal{Z}_K = \{\mathbf{z}^{(k)}\}_{k=1}^{K} \sim p_\theta(\cdot|\mathbf{c})$
    **for** each $\mathbf{z}^{(k)} \in \mathcal{Z}_K$ **do**
        Compute geometric score: $s_k \leftarrow \text{IoU}(\mathbf{z}^{(k)}, \mathbf{z}_{gt})$
        // Soft Rejection Sampling
        **if** $\tau_{valid} \leq s_k < \tau_{match}$ **then**
            Add diverse valid sample: $\mathcal{D}_{train} \leftarrow \mathcal{D}_{train} \cup \{(\mathbf{c}, \mathbf{z}^{(k)})\}$
        **end if**
        // Failure-Driven Augmentation
        **if** $\tau_{low} \leq s_k < \tau_{valid}$ **then**
            Render hard negative input: $\tilde{\mathbf{c}} \leftarrow \phi(d(\mathbf{z}^{(k)}))$
            Add robustness pair: $\mathcal{D}_{train} \leftarrow \mathcal{D}_{train} \cup \{(\tilde{\mathbf{c}}, \mathbf{z}_{gt})\}$
        **end if**
    **end for**
**end for**

// Model: Amortization via GIFT
**while** not converged **do**
    Sample batch $\mathcal{B} \sim \mathcal{D}_{train}$
    Update $\theta$ to minimize log-likelihood objective:
    $\mathcal{L}(\theta) = -\frac{1}{|\mathcal{B}|} \sum_{(\mathbf{c}, \mathbf{z}) \in \mathcal{B}} \log p_\theta(\mathbf{z}|\mathbf{c})$
**end while**

---

# C. Ablations

## C.1. Inference-Time Scaling Ablation

Figure 12 extends the inference-time scaling analysis by including GIFT-FAIL alongside SFT, GIFT-REJECT, and the full GIFT model. Across all compute budgets, the full GIFT model remains the strongest configuration, indicating that output diversification and failure-driven robustness are complementary rather than redundant. Notably, GIFT-FAIL alone already yields a substantial gain over the SFT baseline, suggesting that learning from structured near-miss failures is an effective way to improve single-view Image-to-CAD generation.

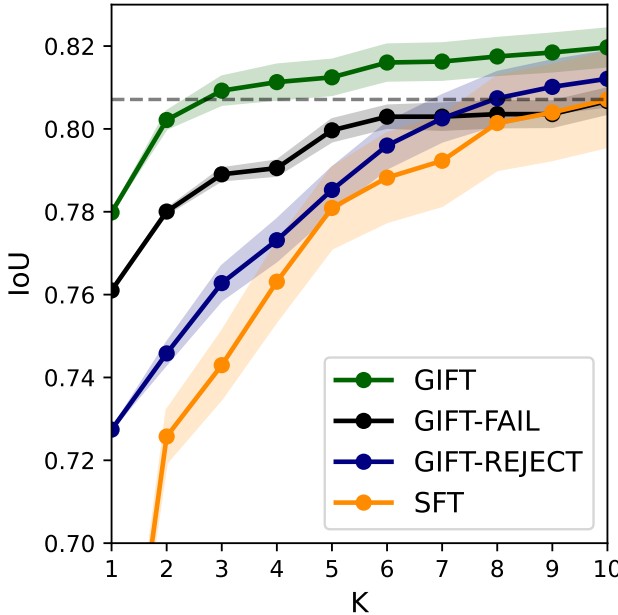

*Figure 12.* Extended inference-time scaling comparison including GIFT-FAIL. Across all compute budgets, GIFT achieves the highest IoU, while GIFT-FAIL and GIFT-REJECT each improve over the SFT baseline. The full GIFT model matches the high-compute performance of SFT with substantially less inference-time sampling.

## C.2. Model Ablation

Table 6 and Figure 13 examine whether the benefits of GIFT depend strongly on the base model used for augmentation and training. Across training phases and sampler choices, the overall trend is consistent: the amortization mechanism itself remains beneficial across model variants. This suggests that GIFT is not tied to a single initialization and can be viewed as a general recipe for converting inference-time geometric feedback into improved supervised training signals.

*Table 6.* Base-model ablation for GIFT. We compare the sampler and training backbone combinations used to generate augmented data and analyze how base-model choice affects downstream performance.

| Base Model | Sampler |
| --- | --- |
| QwenVL-2.5-7B | CAD-Coder-7B |
| QwenVL-2.5-3B | CAD-Coder-7B |
| CAD-Coder-7B | CAD-Coder-7B |

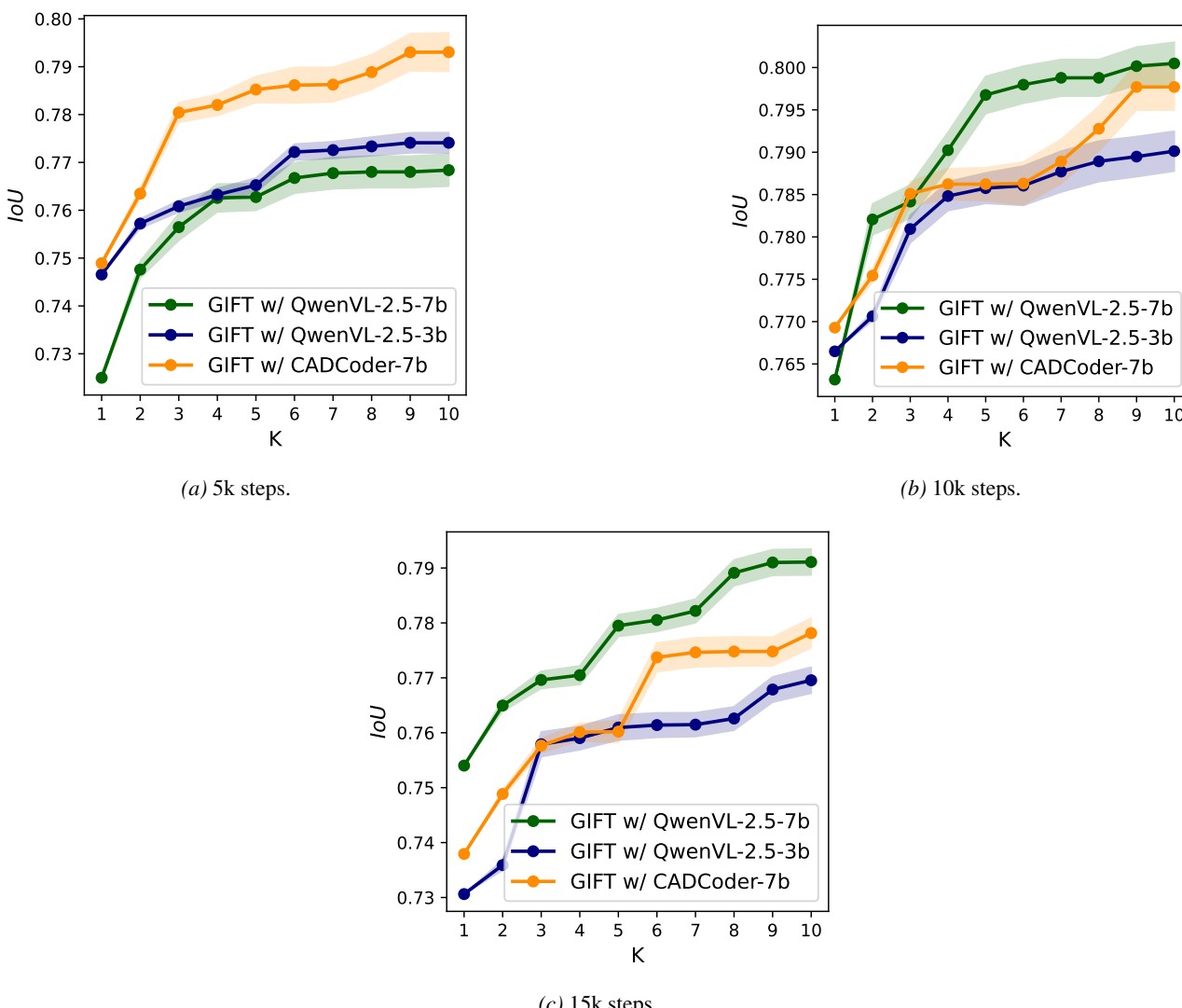

*(a)* 5k steps.

*(b)* 10k steps.

*(c)* 15k steps.

*Figure 13.* Base-model ablation across training phases. We compare GIFT variants bootstrapped with different base and sampler models at 5k, 10k, and 15k training steps. The benefits of GIFT persist across training progress.

## C.3. Sampling Temperature Ablation

Figure 14 studies sensitivity to decoding temperature at inference time. The SFT baseline degrades more noticeably as temperature increases, whereas GIFT maintains stronger performance across the full range, especially in higher-temperature regimes. This supports the claim that GIFT broadens the model's support over valid programs, making it less brittle to exploratory decoding.

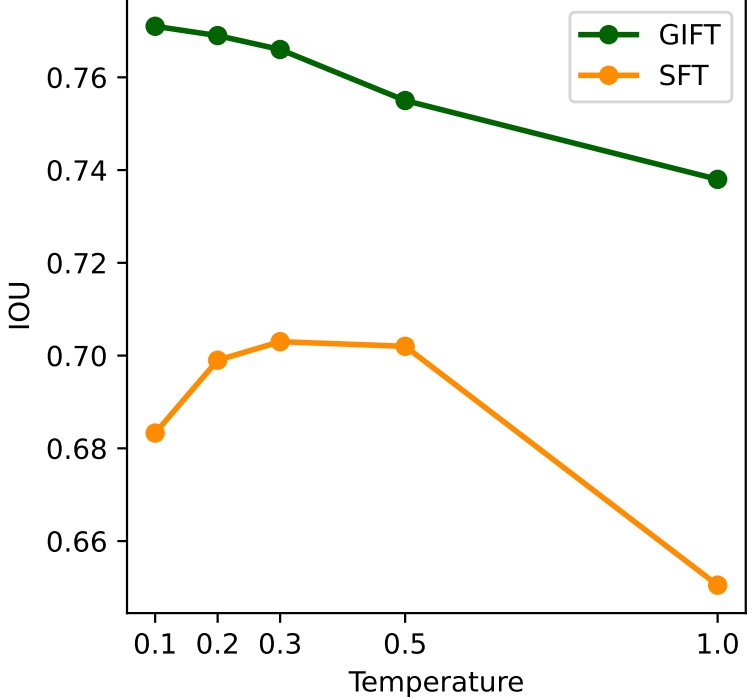

*Figure 14.* Sensitivity to temperature sampling for GIFT and SFT. GIFT is more robust in particular for large temperatures.

# D. Additional Experiments

## D.1. Out-of-Distribution Evaluation

Table 7 provides a more detailed error analysis for both in-distribution and out-of-distribution evaluation. On the in-distribution split, GIFT achieves a higher valid sample rate than SFT while also reducing failed generations and `OpenCASCADE` execution failures. On the out-of-distribution split, the picture is more mixed: GIFT preserves a strong valid-sample rate and lowers direct generation failures, but kernel-related failures increase, indicating that robustness to real-world imagery remains constrained by execution stability as well as prediction quality.

*Table 7.* Valid sample rate and error analysis for GIFT and CAD-Coder-SFT on in-distribution and out-of-distribution test sets. We report pass@1 validity along with failure categories including generation failures, OpenCASCADE failures, and not-solid outputs.

| Method | Split | Valid Sample Rate (%) | Error Analysis (%) | | |
|---|---|---|---|---|---|
| | | pass@1 | Failed Gen | Failed OCC | Not Solids |
| GIFT | in-distribution | 99.06 | 0.20 | 0.38 | 0.35 |
| | out-of-distribution | 96.75 | 0.60 | 2.45 | 0.20 |
| SFT | in-distribution | 98.75 | 0.34 | 0.60 | 0.31 |
| | out-of-distribution | 97.60 | 1.18 | 1.18 | 0.05 |

Figure 15 presents a preliminary out-of-distribution scaling analysis on 400 real-world images (Doris et al., 2025). Although absolute IoU is substantially lower than on the standard test set, GIFT maintains a clear advantage over SFT across compute budgets, suggesting that the benefits of amortized geometric feedback transfer beyond the training distribution. At the same time, the reduced OOD performance highlights that further work is needed to improve visual robustness and compilation reliability under more realistic image conditions.

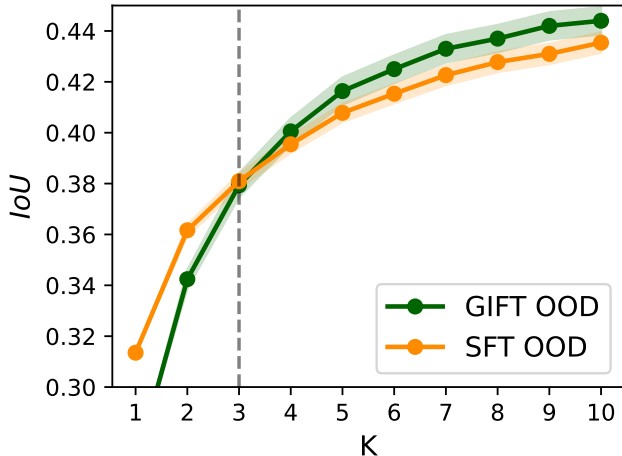

*Figure 15.* Preliminary out-of-distribution evaluation using a set of 400 real-world images, based on the data split provided by (Doris et al., 2025). The results show that GIFT effectively leverages inference-time compute to handle challenging, unseen inputs.

## D.2. Generalization on Fusion360

To evaluate out-of-distribution (OOD) generalization, we benchmark our models on the `Fusion360` test set, which consists of 1,725 challenging, real-world CAD geometries. We compare the baseline `CAD-Coder-SFT` against our proposed `CAD-Coder-GIFT`, both of which were trained exclusively on the GenCAD-Code dataset.

*Table 8.* Failure rates on the Fusion360 test set.

| Model | Valid Sample | Failed Generations | Failed Executions |
|---|---|---|---|
| CAD-Coder-SFT | 96.83% | 1.01% | 1.65% |
| CAD-Coder-GIFT | **97.26%** | **0.56%** | **1.48%** |

**Performance and Inference-Time Scaling**    Our evaluation reveals that incorporating geometric feedback during training provides substantial benefits when generalizing to unseen data distributions. As illustrated in Figure 16a, the GIFT-tuned models achieve a notably higher mean Intersection-over-Union (IoU) compared to the supervised baseline, underscoring their enhanced robustness and output diversity. This performance advantage persists across varying computational budgets during inference. When generating multiple candidate solutions ($K = 10$) for each input, GIFT consistently outperforms CAD-Coder-SFT across the entire test set (Figure 16b). Beyond average metric improvements, we also examined the absolute problem-solving success rate, defined as the proportion of test samples where the model achieves an $IoU > 0.9$. Remarkably, GIFT successfully solves 26% more out-of-distribution problems than the standard SFT approach (Figure 16c). Furthermore, our model size ablations confirm that these algorithmic benefits scale naturally; evaluating the 7B parameter architecture yields further, consistent gains in out-of-distribution generation capabilities (Figure 16f).

**Geometric Complexity and Failure Rate Analysis**    To better understand how these models behave under more challenging topological conditions, we investigated their performance as a function of geometric complexity. Figure 16d highlights the distribution of generated CAD operations, revealing that GIFT possesses a significantly stronger capacity to synthesize longer, more intricate parametric programs without deteriorating into invalid syntax. This capability is further supported by Figure 16e, which demonstrates that GIFT maintains a substantially higher IoU even as the underlying test geometries demand an increasing number of CAD operations. In addition to excelling on complex designs, GIFT exhibits superior structural and syntactic reliability. As detailed in Table 8, the GIFT framework not only yields a higher percentage of geometrically valid, executable samples but also effectively halves the rate of failed generations. It similarly reduces the frequency of execution failures, providing strong evidence that our feedback-tuning mechanism instills a deeper, more resilient understanding of CAD syntax and programmatic constraints.

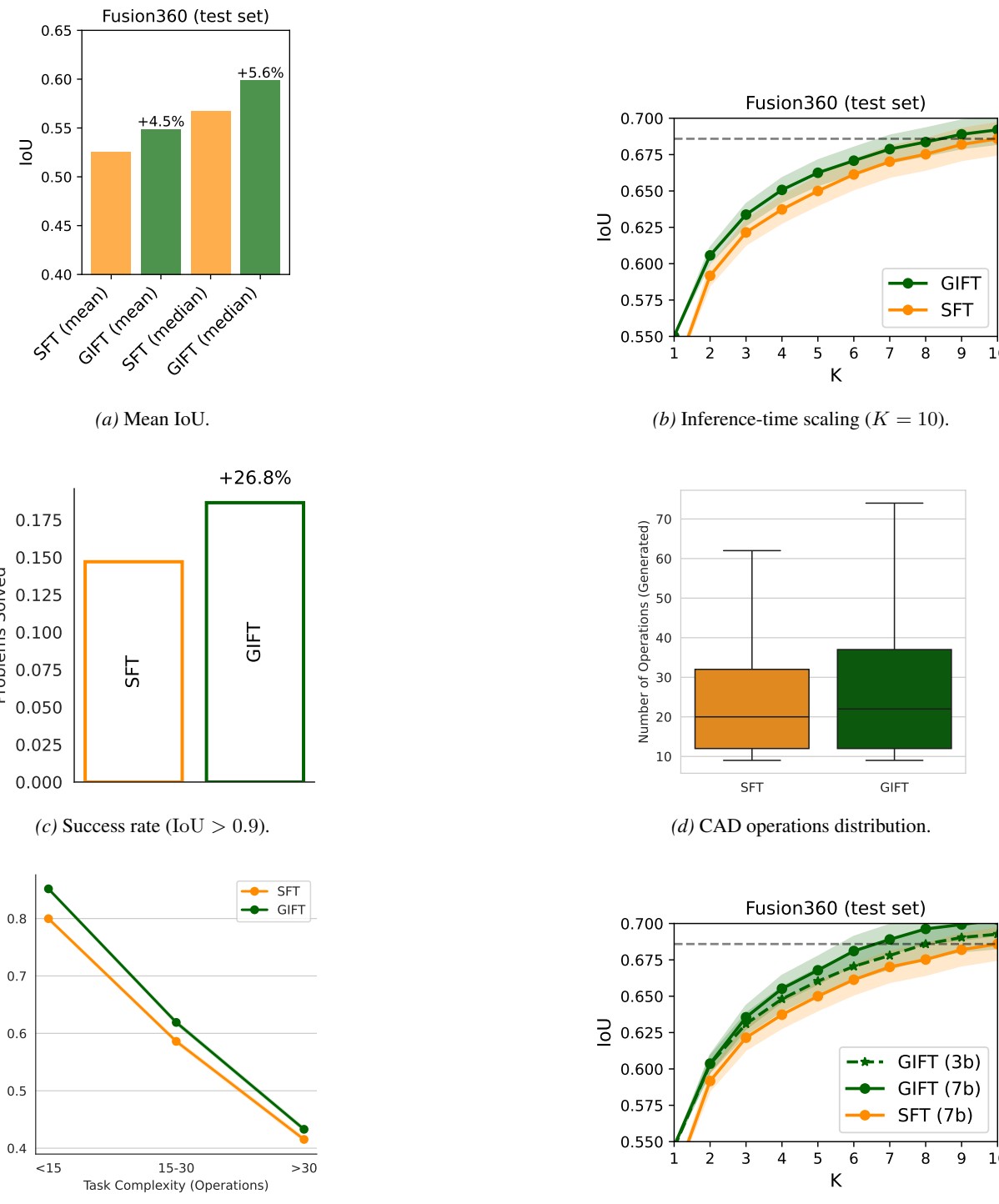

*(a)* Mean IoU.

*(b)* Inference-time scaling ($K = 10$).

*(c)* Success rate ($\mathrm{IoU} > 0.9$).

*(d)* CAD operations distribution.

*(e)* Performance vs. complexity.

*(f)* Model scale ablation.

*Figure 16.* Out-of-distribution evaluation on the Fusion360 test set. The models were evaluated across various dimensions including overall IoU performance (a), compute scaling (b), absolute success rate (c), geometric complexity handling (d, e), and model scale (f).

### D.3. Failure Handling

Table 9 contextualizes our results against stronger multimodal and specialized systems (Kolodiazhnyi et al., 2025), including settings with and without failed generations. The comparison highlights two points. First, failure handling materially affects reported mean IoU, making median IoU a useful complementary statistic for program-generation systems. Second, despite relying only on a single-view image and a general-purpose VLM backbone, CAD-Coder-GIFT closes a meaningful portion of the gap to specialized multimodal pipelines.

*Table 9.* Comparison with recent multimodal and specialized CAD-generation systems. PC denotes point clouds, IMG multi-view images, and IMG-s single-view images; w/ f includes failed generations, while w/o f excludes them. Reporting both mean and median IoU highlights the sensitivity of average performance to failed executions.

|  | Modality | IoU (mean) | IoU (median) |
| --- | --- | --- | --- |
| cadrille-RL (w/o f) | PC | 0.912 | 0.974 |
| cadrille-RL (w/o f) | PC,IMG | 0.920 | 0.978 |
| cadrille-RL (w/ f) | PC | 0.820 | 0.966 |
| cadrille-RL (w/ f) | PC, IMG | 0.827 | 0.972 |
| CAD-Coder-SFT (w/ f) | IMG-s | 0.698 | 0.803 |
| CAD-Coder-GIFT (w/ f) | IMG-s | 0.782 | 0.948 |

### D.4. Performance Metrics and Task Complexity

Figure 17 refines the intrinsic robustness analysis by separating complexity into operation-count and token-count views. Across both measures, GIFT improves median and mean IoU while increasing the fraction of successfully solved problems, especially on more complex examples. This supports the interpretation that GIFT does not merely improve easy cases, but shifts the model toward more reliable geometric reasoning on harder synthesis tasks.

Figure 18 analyzes the distribution of problem complexity in both ground-truth and generated programs. The comparison suggests that the gains from GIFT are not explained simply by producing shorter or structurally simpler programs. Instead, the model continues to generate programs with complexity profiles similar to the target distribution while achieving higher geometric fidelity, indicating better alignment rather than trivial simplification.

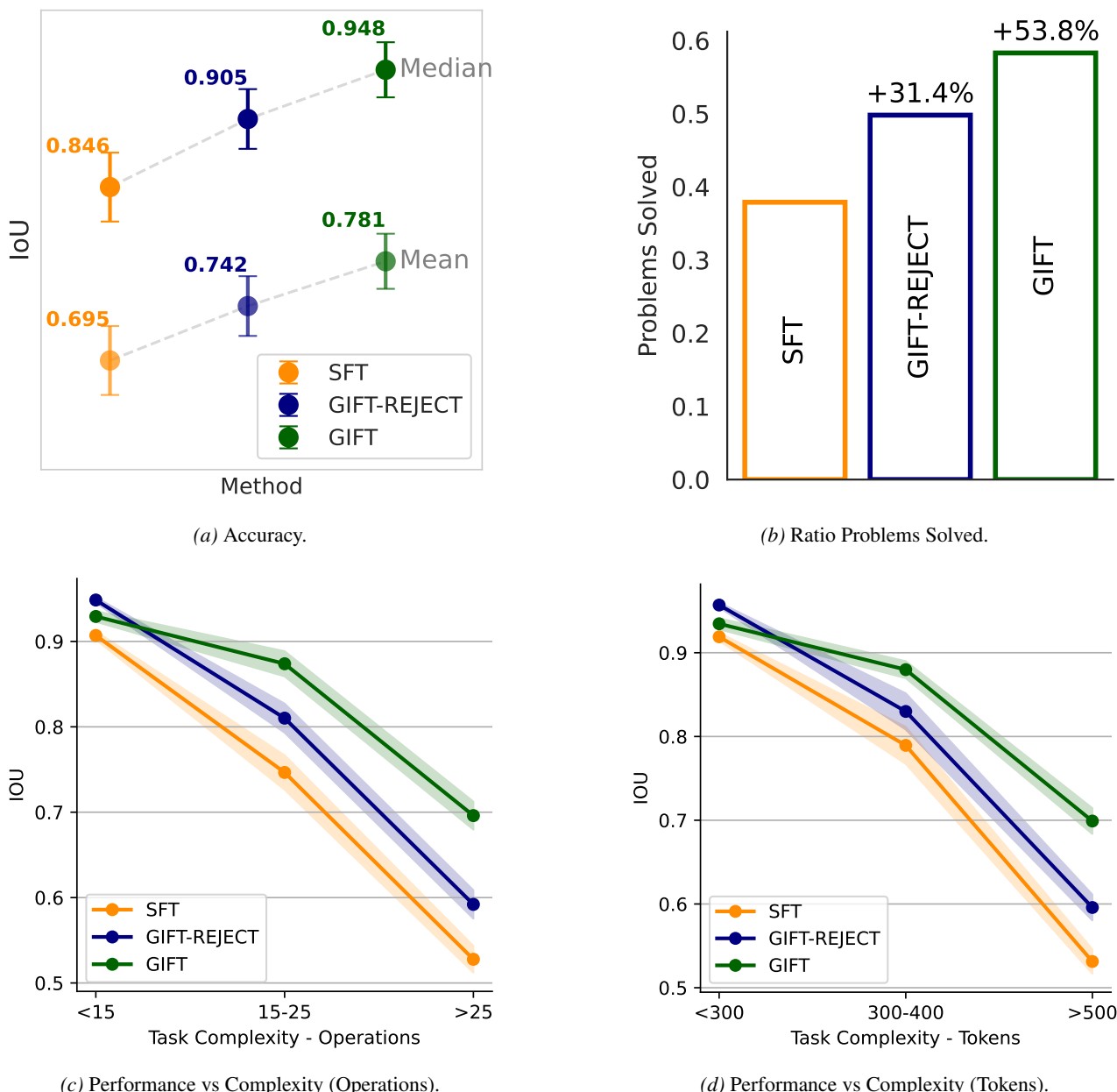

*(a)* Accuracy.

*(b)* Ratio Problems Solved.

*(c)* Performance vs Complexity (Operations).

*(d)* Performance vs Complexity (Tokens).

*Figure 17.* Performance metrics under different data-augmentation strategies. We evaluate Top-K ($K = 10$) generations filtered by geometric validity ($IoU \geq 0.9$ against the ground truth) and compare accuracy, solved-problem rate, and performance as a function of task complexity. GIFT improves both geometric fidelity and robustness, with the largest gains appearing on more complex problems.

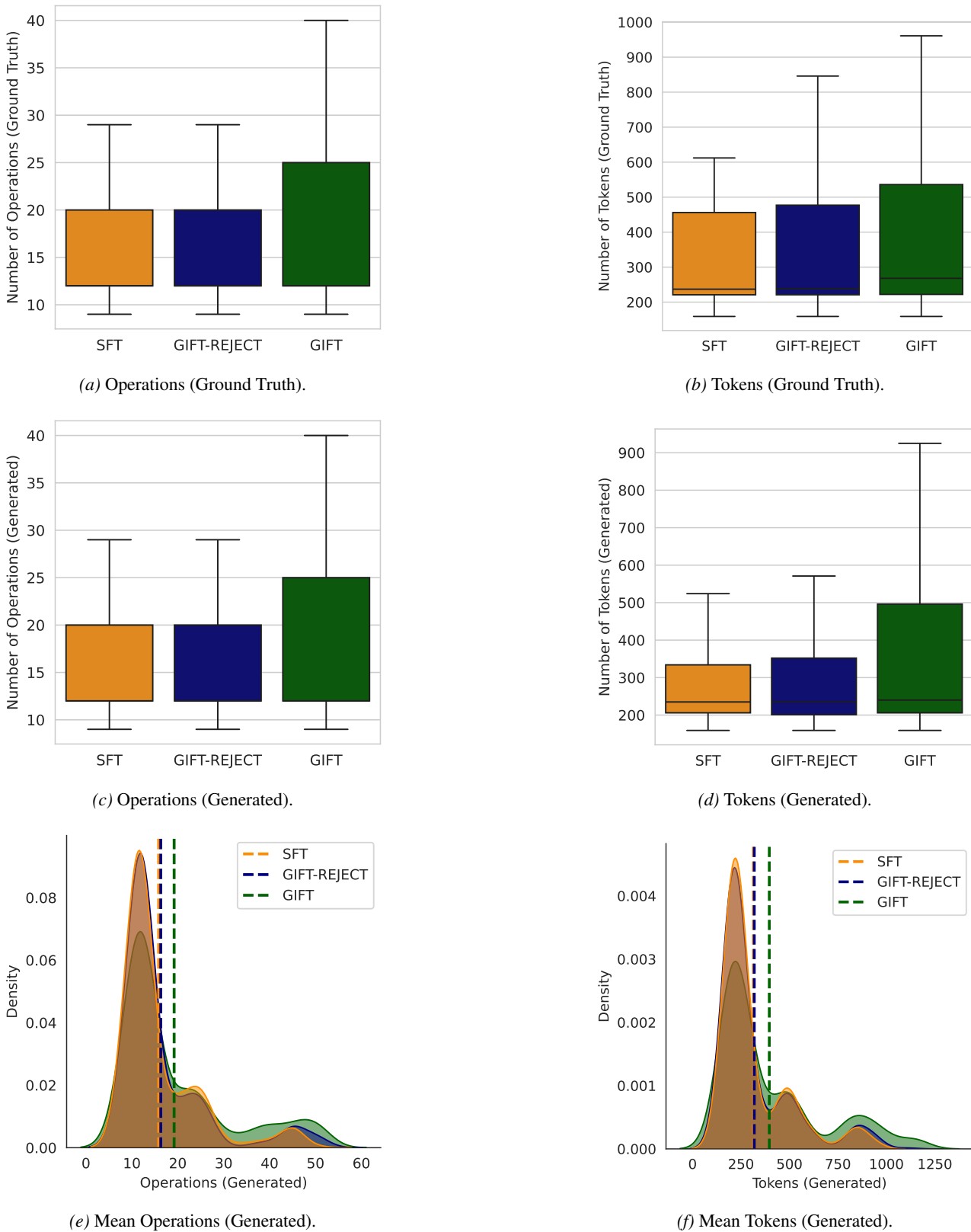

*(a)* Operations (Ground Truth).

*(b)* Tokens (Ground Truth).

*(c)* Operations (Generated).

*(d)* Tokens (Generated).

*(e)* Mean Operations (Generated).

*(f)* Mean Tokens (Generated).

*Figure 18.* Complexity statistics for ground-truth and generated programs with high-fidelity. We compare the distributions of operation counts and token lengths across SFT, GIFT-REJECT, and GIFT. The results show that GIFT improves alignment with valid target structures.

# E. Can Frontier Models Zero-Shot Image-to-CAD?

Table 10 compares GIFT against a frontier general-purpose multimodal model in a zero-shot Image-to-CAD setting. The results show that foundation models exhibit nontrivial geometric competence, especially with higher-resolution inputs, but still trail a specialized domain-adapted system by a large margin. This reinforces the value of targeted geometric supervision and domain-specific amortization even in the era of very capable generalist VLMs.

*Table 10.* Mean and median IoU on a subset of the GenCAD test set for zero-shot Gemini 3.0 Pro and the specialized CAD-Coder-GIFT model. Results are shown for different input resolutions to assess the sensitivity of frontier multimodal models to visual detail.

| Model | Mode | Input Res | IoU (mean) | IoU (median) |
| --- | --- | --- | --- | --- |
| Gemini-3-PRO | Zero-Shot | Low | 0.200 | 0.141 |
| Gemini-3-PRO | Zero-Shot | High | 0.540 | 0.560 |
| CAD-Coder | GIFT | - | 0.782 | 0.948 |

To assess state-of-the-art multimodal capabilities in engineering design, we evaluated Gemini 3.0 Pro (Team et al., 2023) on zero-shot Image-to-CAD generation. As shown in Table 10, performance is heavily dependent on input resolution, with high-resolution inputs more than doubling mean IoU. While this indicates a remarkable emergent geometric understanding in general-purpose foundation models, they still lag behind specialized systems. Our domain-specific CAD-Coder with GIFT significantly outperforms the frontier model, demonstrating that smaller, specialized architectures remain superior for high-fidelity CAD program synthesis.

# F. Dataset Generation

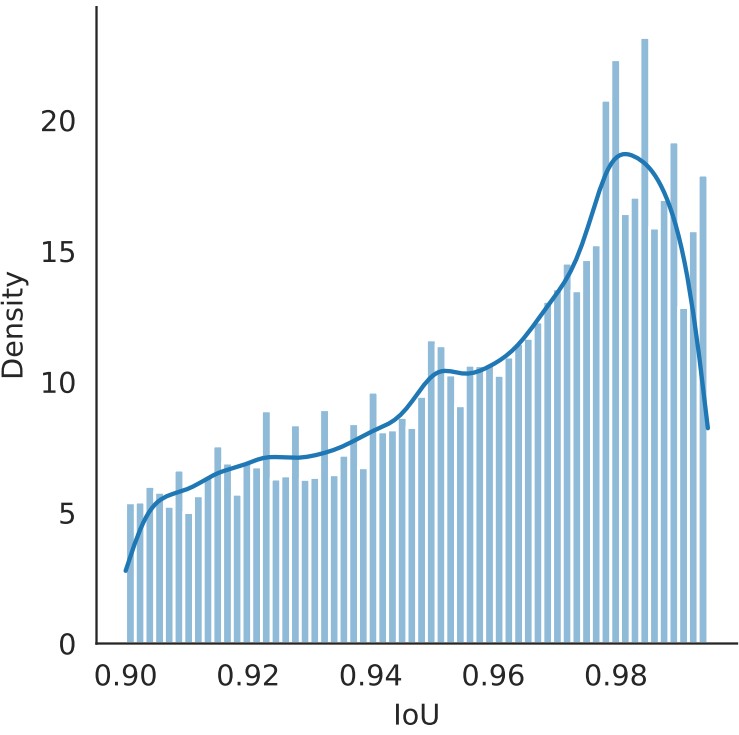

*Figure 19.* Distribution of Top-1 samples with $IoU > 0.9$.

**Sampling Strategy**  Our objective is to curate a diverse dataset of CadQuery programs for single images, effectively mimicking the multiple valid reasoning paths exhibited by large language models. To achieve this, we employ `QwenVL-2.5-7B-CADCoder` as our primary sampler. We strictly utilize the existing training set as source material, leveraging ground truth STEP files as the exclusive source of geometric feedback.The dataset is constructed via a multi-stage pipeline that scales candidate generation across five distinct computational budgets: $N \in \{8, 16, 32, 64, 128\}$.

Starting with a base pool of 80,000 training images, we sample subsets of 10,000 (increasing to 40,000 for the $N = 32$ budget). To maximize semantic diversity, we implement an *inverse scaling* strategy across 29 hyperparameter configurations. As detailed in Table 11, lower compute budgets utilize broader temperature ranges (e.g., $T \in \{0.2, 0.4, 0.6\}$) to encourage exploration, whereas higher budgets prioritize precision through strict, low-temperature sampling ($T = 0.2$).

*Table 11.* Dataset budget mix and sampling hyperparameter configurations used during data generation. The Samples column reports the selected subset size retained for each compute budget after top-10% filtering.

| Budget | Inputs | Samples | $(T, p)$ |
|---|---|---|---|
| $N = 8$ | 10,000 | 10,000 | $T = 0.2 : p \in \{0.7, 0.8, 0.9, 1.0\}$ 
 $T = 0.4 : p \in \{0.7, 0.8, 0.9, 1.0\}$ 
 $T = 0.6 : p \in \{0.8, 0.9, 1.0\}$ |
| $N = 16$ | 10,000 | 20,000 | $T = 0.2 : p \in \{0.7, 0.8, 0.9, 1.0\}$ 
 $T = 0.4 : p \in \{0.8, 0.9, 1.0\}$ |
| $N = 32$ | 40,000 | 120,000 | $T = 0.2 : p \in \{0.7, 0.8, 0.9, 1.0\}$ 
 $T = 0.4 : p \in \{0.9, 1.0\}$ |
| $N = 64$ | 10,000 | 60,000 | $T = 0.2 : p \in \{0.8, 0.9, 1.0\}$ |
| $N = 128$ | 10,000 | 120,000 | $T = 0.2 : p \in \{0.9, 1.0\}$ |

**Selection Strategy**   From approximately 1 million generated raw samples, we compute the Intersection over Union (IoU) against ground truth scripts to partition the data into three distinct splits. The FULL split retains the top 10% of successful generations per input (where $IoU > 0.5$). The FAIL split isolates failure modes by selecting the highest-scoring candidate for inputs where satisfactory reconstruction failed ($IoU < 0.9$).

Finally, we construct the Soft Rejection Sampling (SRS) set for high-quality fine-tuning. This split combines the original ground truth data with the single best generated candidate (Top-1) falling within the range $0.9 < IoU < 0.99$ (excluding exact duplicates).

**Oversampling**   While selecting the best of $K$ generations is generally effective, it relies on the assumption that a valid candidate exists within the sample pool. Our analysis indicates that for a significant portion of the training set, simply increasing sampling frequency fails to yield high-IoU results. To mitigate this, we leverage geometric feedback to identify these persistent failures and selectively augment the dataset with these hard samples.

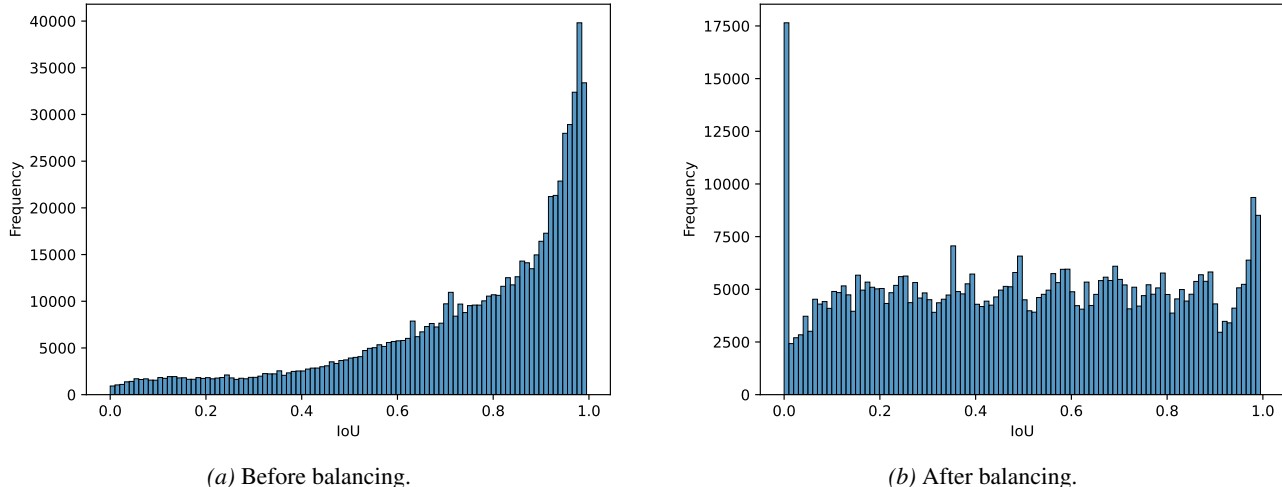

*(a)* Before balancing.          *(b)* After balancing.

*Figure 20.* IoU distribution of sampled programs before and after balancing. Balancing reduces over-representation of the densest score ranges and produces a more even distribution of training examples across quality levels, improving the usefulness of both SRS and FDA data.

# G. Amortizing Inference-Time Augmentation: Gradients

In this section, we analyze the gradients of the GIFT objective function and establish the theoretical connection between our rejection-sampling approach and standard policy gradient methods.

**The Reinforcement Learning Perspective**  Ideally, we aim to maximize the expected geometric validity (IoU) of the generated programs. Let $J(\theta)$ be the expected reward under the policy $p_\theta$:

$$J(\theta) = \mathbb{E}_{\mathbf{c} \sim q_\mathcal{D}, \mathbf{z} \sim p_\theta(\cdot | \mathbf{c})}[r(\mathbf{z}, \mathbf{z}_{gt})], \tag{6}$$

where the reward $r(\mathbf{z}, \mathbf{z}_{gt}) = f(\mathbf{z})$ is the Intersection over Union (IoU) defined in the Method. Maximizing this objective via gradient ascent yields the standard REINFORCE policy gradient:

$$\nabla_\theta J(\theta) = \mathbb{E}_{\mathbf{c}, \mathbf{z}} \left[ f(\mathbf{z}) \nabla_\theta \log p_\theta(\mathbf{z} | \mathbf{c}) \right]. \tag{7}$$

In standard online RL (e.g., PPO), this expectation is approximated by sampling batches from the *current* policy $\pi_\theta$ during training. However, this is computationally prohibitive for CAD synthesis because computing $f(\mathbf{z})$ requires executing an expensive geometric kernel for every step.

## G.1. The GIFT Approximation

GIFT bypasses online sampling by approximating the expectation using a static, high-quality dataset collected via Inference-Time Scaling (ITS). We treat the sample selection process as an importance sampling step where the weights are binary (hard rejection).

**SRS Gradient (Diversity)**  The SRS component effectively performs Reward-Weighted Regression with a hard threshold. Instead of weighting every sample by its scalar reward $f(\mathbf{z})$, we define a binary weight $w_{srs}(\mathbf{z}) = \mathbb{1}[\tau_{valid} \leq f(\mathbf{z}) < \tau_{match}]$.

The gradient for the SRS objective provided in Eq. 4 is:

$$\nabla_\theta \mathcal{F}_{SRS} = \mathbb{E}_{(\mathbf{c}, \mathbf{z}) \sim \mathcal{D}_{SRS}} \left[ \nabla_\theta \log p_\theta(\mathbf{z} | \mathbf{c}) \right]. \tag{8}$$

Substituting the definition of $\mathcal{D}_{SRS}$, this is equivalent to a Monte-Carlo approximation of the policy gradient where samples with low reward are zeroed out:

$$\nabla_\theta \mathcal{F}_{SRS} \approx \sum_{n=1}^{N} \sum_{k=1}^{K} w_{srs}(\mathbf{z}^{(k)}) \nabla_\theta \log p_\theta(\mathbf{z}^{(k)} | \mathbf{c}^{(n)}). \tag{9}$$

This formulation stabilizes training by treating the generated programs $\mathbf{z}^{(k)}$ as fixed "pseudo-ground-truths," converting the unstable RL problem into a standard supervised maximum likelihood estimation (MLE) problem.

**FDA Gradient (Robustness)**  The FDA component introduces a distinct gradient signal. Unlike SRS, which optimizes the likelihood of the generated sample $\mathbf{z}$, FDA optimizes the likelihood of the *ground truth* $\mathbf{z}_{gt}$ conditioned on a noisy input $\tilde{\mathbf{c}}$. The objective is a supervised denoising loss:

$$\mathcal{F}_{FDA} = \mathbb{E}_{(\tilde{\mathbf{c}}, \mathbf{z}_{gt}) \sim \mathcal{D}_{FDA}}[\log p_\theta(\mathbf{z}_{gt} | \tilde{\mathbf{c}})]. \tag{10}$$

The gradient is straightforward:

$$\nabla_\theta \mathcal{F}_{FDA} = \sum_{n=1}^{N} \sum_{k=1}^{K} w_{fda}(\mathbf{z}^{(k)}) \nabla_\theta \log p_\theta(\mathbf{z}_{gt}^{(n)} | \phi(d(\mathtt{sg}[\mathbf{z}^{(k)}]))). \tag{11}$$

Here, the gradient signal does not encourage the production of the failed sample $\mathbf{z}^{(k)}$. Instead, it updates the visual encoder and language decoder to be invariant to the geometric perturbations present in $\tilde{\mathbf{c}} = \phi(d(\mathbf{z}^{(k)}))$, pulling the distribution back towards the mode $\mathbf{z}_{gt}$.

**Augmented Gradient Update**    Combining the Base (SFT), Diversity (SRS), and Robustness (FDA) terms, the final parameter update at each step is:

$$\Delta\theta \propto \underbrace{\sum_n \nabla_\theta \log p(\mathbf{z}_{gt}^{(n)}|\mathbf{c}^{(n)})}_{\text{SFT: Grounding}} + \underbrace{\sum_{n,k} w_{srs}^{(k)} \nabla_\theta \log p(\mathbf{z}^{(k)}|\mathbf{c}^{(n)})}_{\text{SRS: Diversity}} + \underbrace{\sum_{n,k} w_{fda}^{(k)} \nabla_\theta \log p(\mathbf{z}_{gt}^{(n)}|\tilde{\mathbf{c}}^{(k)})}_{\text{FDA: Robustness}} \tag{12}$$

This formulation allows GIFT to amortize the cost of search into the model weights offline, avoiding the variance and computational cost of online policy gradient estimation.

# H. Visualizations

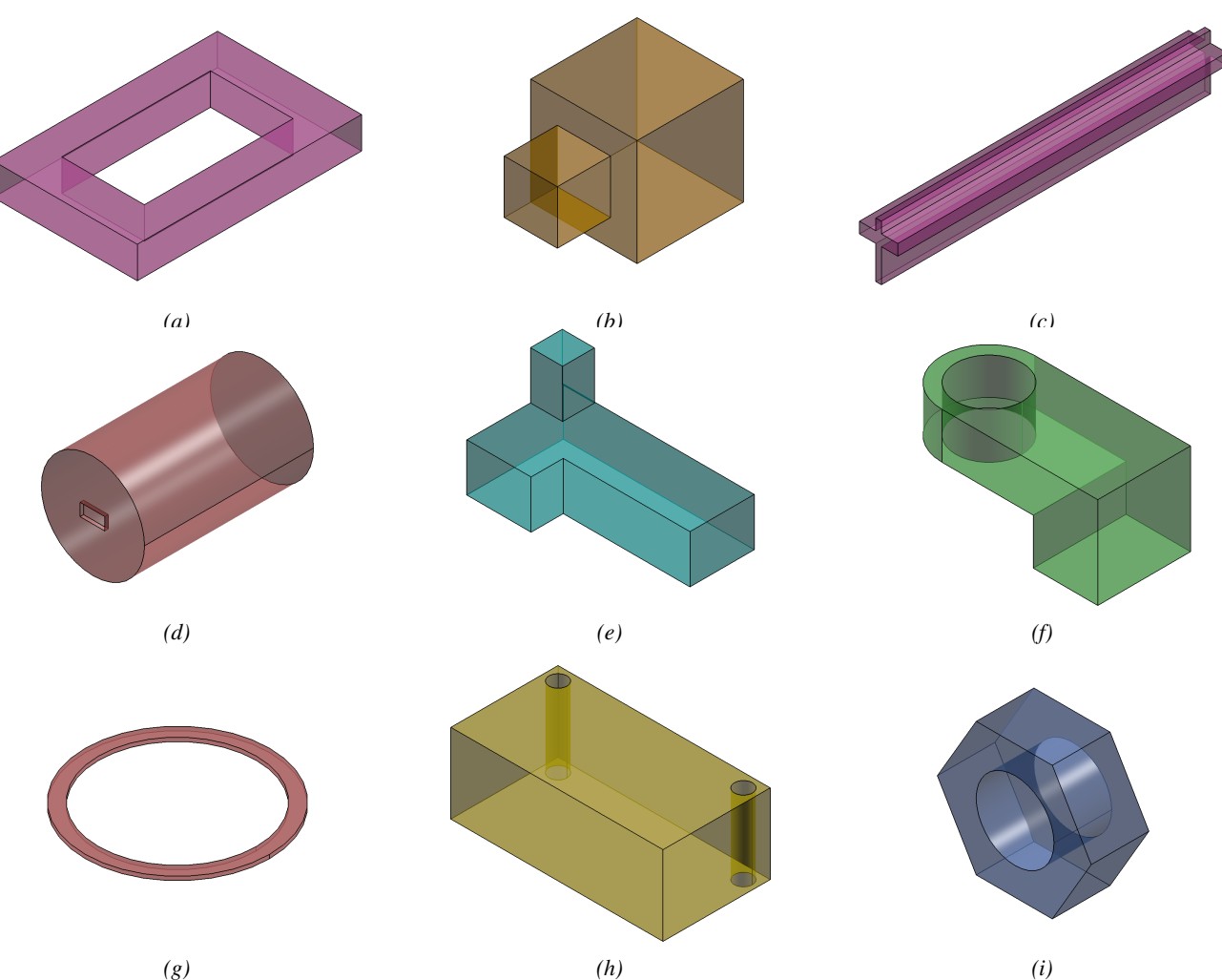

*Figure 21.* Rendered examples of generated CAD programs with intermediate IoU quality ($IoU > 0.5$ and $IoU < 0.9$). These visualizations show how geometrically imperfect but executable programs can be projected back into the image domain and reused as synthetic inputs for Failure-Driven Augmentation.

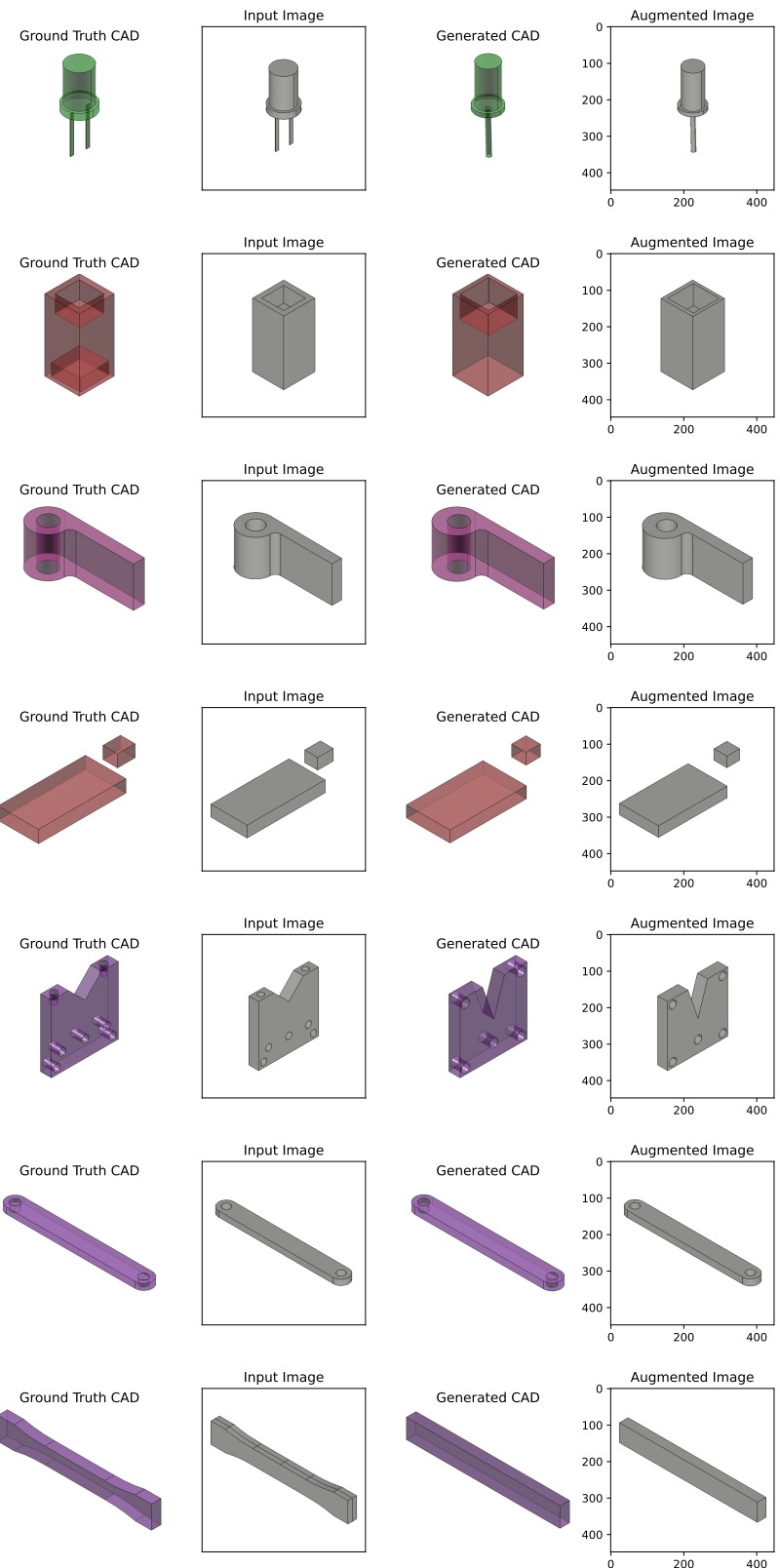

*Figure 22.* Examples of low-IoU but executable generations used for Failure-Driven Augmentation. The rendered failures provide structured hard negatives that train the model to recover the original ground-truth program from imperfect visual geometry.

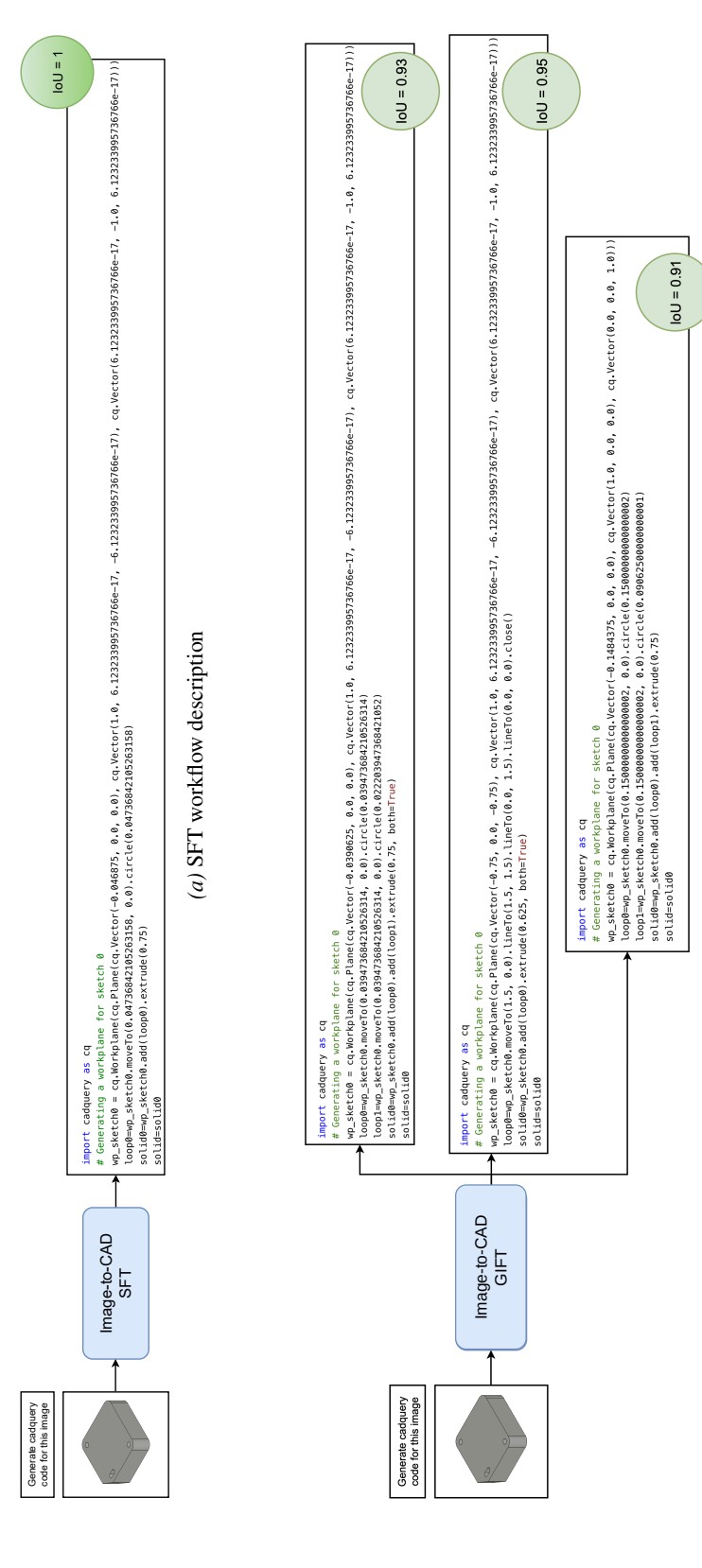

*Figure 23.* Workflow comparison between standard SFT and GIFT. In the SFT pipeline, each input image is paired with a single ground-truth program. In GIFT, multiple sampled programs are geometrically verified, and both high-quality alternatives and structured failures are converted into additional supervised training signals.

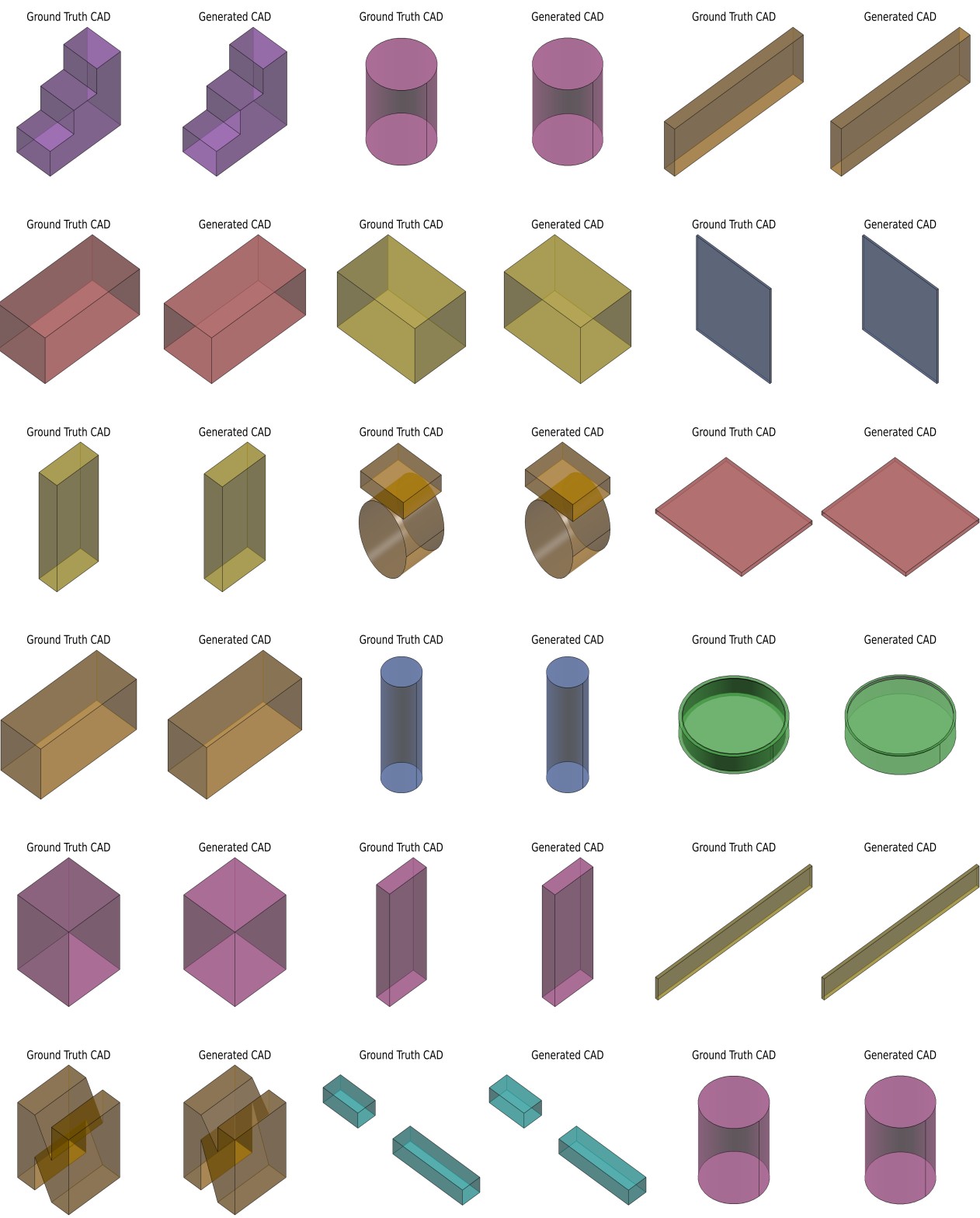

*Figure 24.* Example of high IoU CAD generations used for SRS augmentation.

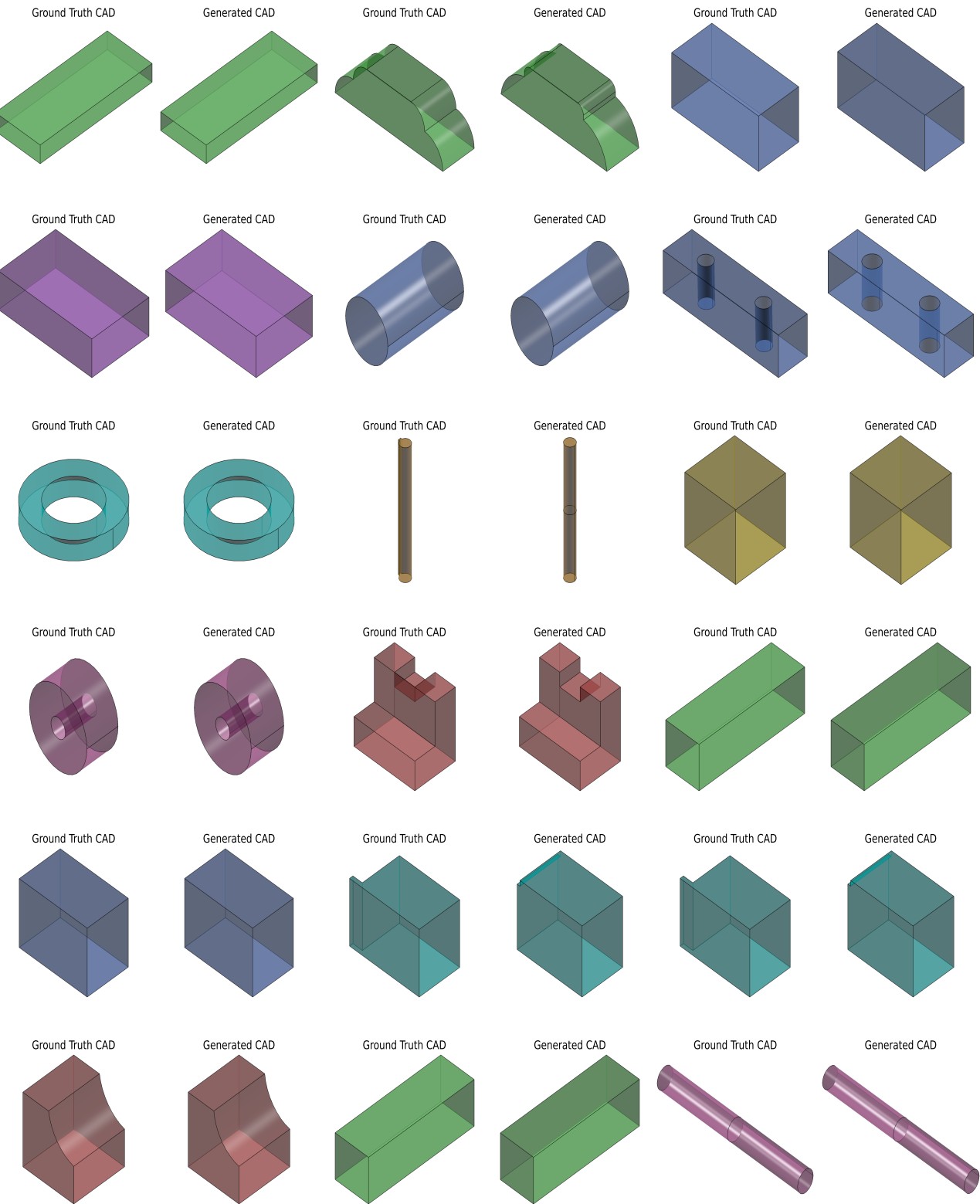

*Figure 25.* Example of high IoU CAD generations used for SRS augmentation.

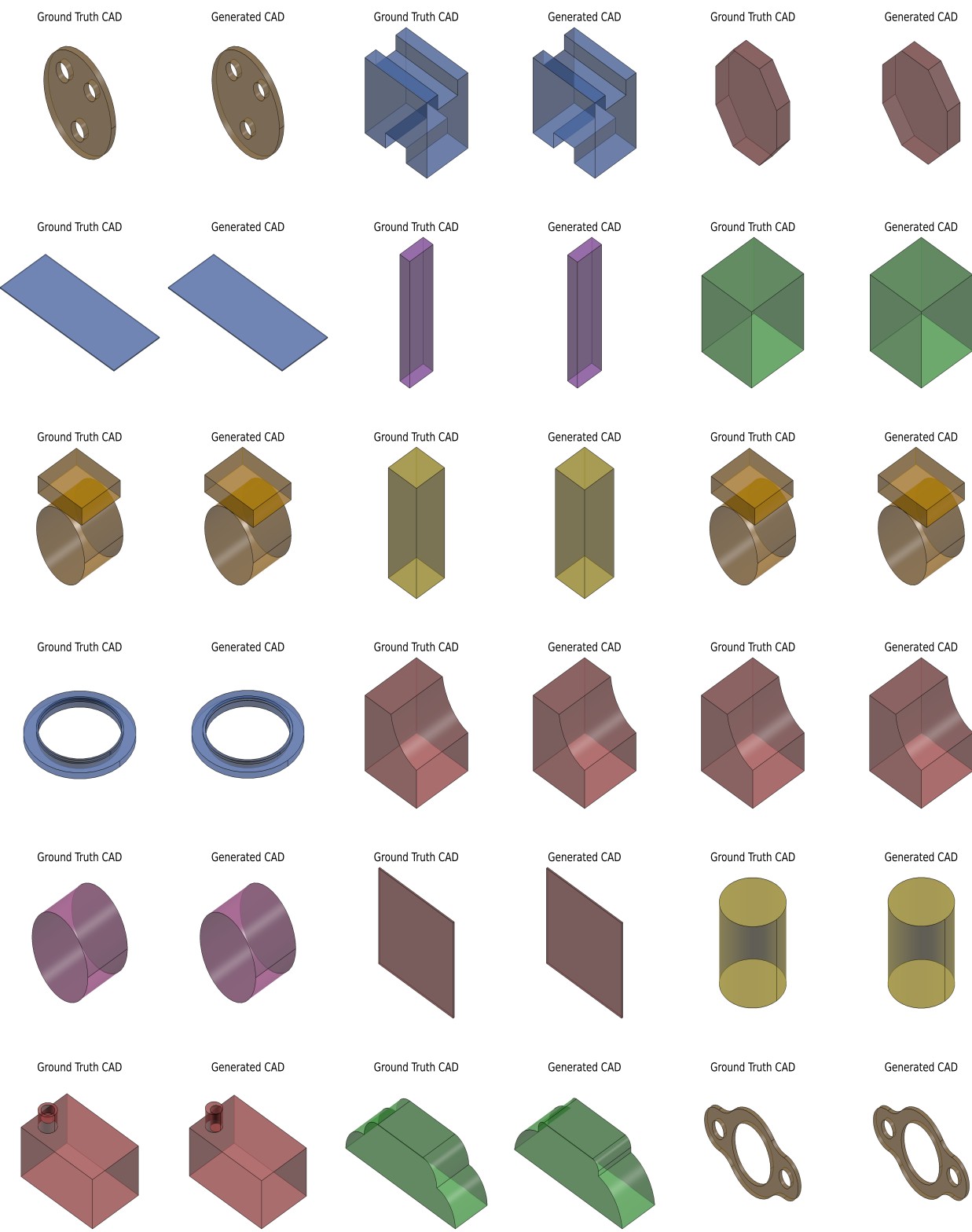

*Figure 26.* Example of high IoU CAD generations used for SRS augmentation.

# I. Details

*Table 12.* Experimental setup and implementation details for training, inference-time scaling, and GIFT-specific augmentation. We summarize model backbones, dataset sizes, sampling budgets, filtering thresholds, optimization settings, and hardware.

| Hyperparameter | Value |
|---|---|
| *Model Architecture* | |
| Base Model | Qwen-VL-2.5-7B-Instruct/Qwen-VL-2.5-3B-Instruct |
| Parameter Count | 7B/3B |
| *Datasets* | |
| Source Dataset | GenCAD-Code / DeepCAD |
| Training Size (Original SFT) | 163k |
| Augmented Dataset Size | 370k |
| IN Test Set Size | 7355 |
| OOD Test Set Size | 400 |
| *Inference-Time Scaling (Data Generation)* | |
| Sampling Budgets ($N$) | $\{8, 16, 32, 64, 128\}$ |
| Sampling Temperatures ($T$) | $\{0.2, 0.4, 0.6\}$ |
| Top-$p$ values | $\{0.7, 0.8, 0.9, 1.0\}$ |
| Sampler Model | QwenVL-2.5-7B-CADCoder |
| *GIFT Configuration* | |
| SRS Validity Range ($\tau_{valid}$) | $0.9 \leq \text{IoU} < 0.99$ |
| FDA Range ($\tau_{noise}$) | $0.5 \leq \text{IoU} < 0.9$ |
| Geometric Kernel | OpenCASCADE / CadQuery |
| *Training Optimization* | |
| Optimizer | AdamW |
| Learning Rate | $2e-5$ |
| LR Scheduler | Cosine Decay |
| Batch Size / GPU | 4 |
| Training Epochs | 10 |
| Precision | BF16 |
| Hardware | 8x A100 80GB |

