# OpenReview forum: "GIFT: Bootstrapping Image-to-CAD Program Synthesis via Geometric Feedback"
_ICML.cc/2026/Conference — ICML 2026 regular_

### Official Review · Reviewer_QNNZ · 2026-02-19

**Soundness:** 3
**Presentation:** 3
**Significance:** 3
**Originality:** 3
**Overall Recommendation:** 4
**Confidence:** 3

**Summary:**

The paper is about image-to-code (cadquery) generation in the CAD manufacturing domain. The focus is how to improve alignment between image geometry and code. Without RL, the authors propose a bootstrapping framework with 2 tricks (SRS + FDA) to expand the SFT training set, using IoU from a CAD kernel as the key verifier.

**Compliance With Llm Reviewing Policy:**

Affirmed.

**Final Justification:**

Thank you. Based on the current version and the rebuttal results, I'll maintain my positive rating.

**Key Questions For Authors:**

1. In Figure 1, CAD-Coder-SFT baseline should correspond to the grey dashed line, not blue?

2. Some figures are shown but not clearly referenced in the main text (e.g., Figure 1).

3. the structure of the Method section could be improved. Currently only Section 3.1 is numbered, without 3.2, etc, which feels uneven. I suggest restructuring for clarity. For example, shift *Inference-Time Scaling for CAD Generation* and *Bootstrapping* for better logical flow.

4. Table 5 and Table 4 appear in a reversed order.

5. In Fig.12 caption, the sentence:
“GIFT outperforms both SFT, GIFT-FAIL, and GIFT-REJECT…”
is confusing because GIFT-FAIL and GIFT-REJECT are also geometric-feedback-based methods.

5. In Fig.20, subfigure captions (a)(b)(c) are occluded.

**Limitations:**

yes

**Strengths And Weaknesses:**

Strengths:

From the experimental results, the method is clearly strong and well-engineered, and the improvements are convincing.

Weaknesses:

1. In SRS, it uses 0.9 ≤ IoU < 0.99 as *high-quality*. My concern is: what is the evidence that for example 0.9 corresponds to semantic correctness? could there still be missing features (e.g., a hole slightly misplaced)? The rationale for choosing 0.9 seems mainly empirical and distribution-based (3.1, threshold selection), and Figure 9(b), from my understanding, a turning point in sample number does not necessarily imply a **sample quality improvement in geometry**.
The same issue applies to FDA. Is 0.51 fundamentally the same as 0.89 in terms of semantic consistency?

2. In FDA, the model sees a rendered failure (image) and is trained to predict the GT code. could FDA introduce semantic distortion? The authors describe this as denoising, but I worry about a possible case: suppose the GT is a cube with a hole. The model generates a wrong program (IoU 0.6) that produces a solid cube (forgot the hole). FDA renders this solid cube as input, and uses the *cube with hole* code as label. So the model is trained to learn: when you see a solid cube, please imagine a hole and generate it.

3. Samples with IoU < 0.5 are discarded. Is there potential to handle them differently, e.g., with RL? since most case studies (e.g., Figure 20) seem to focus on relatively reasonable examples.

4. If we had a diverse and low-noise dataset, would SRS&FDA still provide substantial gains? In other words, is GIFT’s effectiveness partly dependent on the limitations of the dataset?

---

> ### Author Rebuttal · Authors · 2026-03-26
>
> **Reviewer:**
>
> > In SRS, it uses 0.9 ≤ IoU < 0.99 as high-quality. My concern is: what is the evidence that for example 0.9 corresponds to semantic correctness? The rationale for choosing 0.9 seems mainly empirical and distribution-based
>
> **Authors:**
>
> You are right: the threshold are chosen in a data-driven manner leveraging inference compute. This is the core idea of GIFT as an augmentation strategy. We do not claim that IoU ≥ 0.9 guarantees semantic equivalence. We use 0.9 as a conservative operating point that trades geometric fidelity against program diversity, and we will revise the paper to frame 0.9/0.5 as empirically chosen thresholds rather than universal semantic cutoffs.
>
> - For SRS, setting the threshold to 0.9 preserves geometric quality while still allowing meaningful variation in program form. Empirically the model can solve more and harder problems (Fig 2/16/17).
>
> - For FDA, the 0.5 ≤ IoU < 0.9 range is broad by design. It serves to capture geometrically plausible "near-miss" programs that encode structured errors rather than random noise. We agree that 0.51 and 0.89 are not semantically equivalent. In the current paper we group them because both are recoverable near-misses, but this is a simplification; a finer-grained weighting by IoU or failure type is a sensible extension.
>
> -------
>
> **Reviewer:**
>
> > could FDA introduce semantic distortion?
>
> **Authors:**
>
> We agree this is the central risk of FDA. **Our claim is not that the rendered failure is semantically correct, but that sufficiently close executable near-misses can provide useful corrective supervision**. The new random-augmentation control is relevant here: if the gain came mainly from generic denoising or semantic drift, standard image perturbations would help similarly, which they do not (See `xLhT`).
>
> - FDA specifically uses the model's own generated executable code to construct a challenging input.
>
> - If the model originally missed a hole, FDA renders that failed visual state (the solid cube) and pairs it with the correct ground-truth code.
>
> - Rather than teaching the model to randomly hallucinate missing features, this process provides a targeted corrective gradient signal. It trains the visual encoder and language decoder to become invariant to the specific geometric errors present in the rendered failure, effectively pulling the output distribution back toward the correct mode.
>
> - While standard random augmentations are widely used to improve generalization in vision-language models, FDA adapts this proven principle for engineering. Instead of applying arbitrary noise, FDA leverages the geometric structure of CAD to provide targeted feedback that directly addresses the model's actual weaknesses.
>
> -------------
>
> **Reviewer:**
>
> > Samples with IoU < 0.5 are discarded.
>
> **Authors:**
>
> While the full [0, 1] IoU range could be used to build an advantage for online RL, samples with IoU < 0.5 (roughly the bottom 10%) are typically degenerate or non-executable. These are "easy negative" with little information for a post-trained model. Since GIFT is a data augmentation framework, discarding these catastrophic failures ensures our dataset only contains high-quality diversity (SRS) and structured, recoverable errors (FDA).
>
> --------
>
> **Reviewer:**
>
> > If we had a diverse and low-noise dataset, would SRS&FDA still provide substantial gains? In other words, is GIFT’s effectiveness partly dependent on the limitations of the dataset?
>
> **Authors**:
>
> We have not yet tested GIFT on a substantially cleaner or more diverse dataset, so we do not want to overclaim. Our hypothesis is that the benefit would persist because GIFT is driven by model errors rather than only by dataset noise, but we will present that as a hypothesis rather than a demonstrated fact.
>
> ---
>
> ### **Formatting and Adjustments**
>
> **Authors**:
>
> Thanks for the feedback. Here the list of changes we made:
>
> - Figure 1: We corrected the caption typo to accurately reflect the SFT baseline as the grey dashed line. We also added an early, explicit reference to Figure 1 in the Introduction to properly anchor it in the text.
>
> - Method: We restructured Section 3 for a more balanced logical flow. We added proper subsection numbering (e.g., 3.2) and shifted the "Inference-Time Scaling for CAD Generation" and "Bootstrapping" paragraphs to better introduce the core pipeline before detailing the specific augmentation strategies.
>
> - Ordering: We adjusted the placement of Table 4 and Table 5 so they appear sequentially and align properly with their callouts in the text.
>
> - Figure 12: We revised this caption to be much more precise. Our intent was to emphasize that the full GIFT framework is a stronger approach than using either of its individual components alone.
>
> - Figure 20: We adjusted the bounding boxes and image scaling in Figure 20 so that subfigure captions (a), (b), and (c) are fully visible and no longer occluded by the CAD renders.

---

> > ### Author Rebuttal · Reviewer_QNNZ · 2026-04-01
> >
> > It doesn't really solve my core concerns, but reframes the method’s design choices as empirical points. It does not **provide evidence** that:
> >
> > - the thresholds meaningfully correspond to semantic correctness or recoverability (without this, I'm afraid the choice of this threshold has to be justified by exhaustive ablations),
> > - the method remains effective beyond the current dataset. This limits its guidance value for the AI community.

---

> > > ### Author Response · Authors · 2026-04-04
> > >
> > > ## Ablation Experiments
> > >
> > > ## https://anonymous.4open.science/r/gift-rebuttal-F818/README.md
> > >
> > > -------
> > >
> > > **Reviewer**:
> > >
> > > > the choice of this threshold has to be justified by exhaustive ablation
> > >
> > > **Authors**:
> > >
> > > We agree that ablating the threshold is a valuable addition to our work. While we cannot complete a full sweep within the short rebuttal period, **we commit to including one in the camera-ready version**.
> > >
> > > In the meantime, we provide preliminary results for GIFT-FAILURE using SRS with thresholds in [0.3, 0.5, 0.7] (see `Ablation 11` in the anonymized repo). We are using a smaller model (3b) for tuning and shorter runs. Please note that `Appendix C` already contains our initial model and temperature ablations.
> > >
> > > See also `Ablation 8 and 9` in the anonymized repo - where we show the **overlap at different IoU thresholds for simple shapes**. As you can see, IoU >= 0.9 overlaps the shapes by a large degree.
> > >
> > > Regarding correctness and recoverability, we believe that our **empirical results demonstrate that SRS and FDA augmentations consistently improve both in-distribution and out-of-distribution performance** (as `Ablation 1 to 7` in the anonymized repo). In particular:
> > >
> > > - `Figures 2b, 2c, and 2d`, along with `Figures 16 and 17 in Appendix D`, illustrate that GIFT successfully solves a **higher fraction of problems**, including **more complex ones with greater token lengths and CAD operations**.
> > >
> > > - `Figure 11` highlights **improvements across standard text similarity metrics** (BLEU, ROUGE, and METEOR) when comparing our generated code to the ground truth. Together, these findings strongly suggest that our task- and model-dependent thresholds - derived via inference compute - are highly effective.
> > >
> > > - We also report an additional text similarity metric (CIDEr score - `Ablation 10` in the anonymized repo), where **GIFT improves by 30% over SFT**, showing that the augmentation strategies employed in GIFT help the model generalize and learn the conditional distribution of programs.
> > >
> > > -----------
> > >
> > > **Reviewer**:
> > >
> > > > the method remains effective beyond the current dataset.
> > >
> > > **Authors**:
> > >
> > > Thank you for raising this point. We provide out-of-distribution results on `Fusion360 test set` in the anonymized repo (`Ablation 1 to 7` ).
> > >
> > > We consider CAD-Coder-SFT and a smaller version of CAD-Coder-GIFT (3b variant) to speed up sampling and evaluation.
> > >
> > > We are working on extending these results. In particular:
> > >
> > > 1) `IoU (mean and median) performance [Fusion360]`
> > >      - GIFT improves mean and median IoU performance over the baseline in an out-of-distribution scenario.
> > >
> > > 2) `Inference-Time Scaling curves [Fusion360]`
> > >     - GIFT outperforms the SFT baseline for all budgets K=[1,2,3,4,5,6,7,8,9,10].
> > >
> > > 3) `Fraction of Problems Solved [Fusion360]`
> > >      - GIFT solves 26 % more problems with high IoU.
> > >
> > > 4) `Number of Generated CAD operations [Fusion360]`
> > >
> > >     - Qualitative result that maps to the results in Figure 2 and Appendix D - Figure 17.
> > >
> > > 5) `IoU vs Generated Program Complexity [Fusion360]`
> > >
> > >     - GIFT consistently solves more complex problems.
> > >
> > > 6) `Failure Rates [Fusion360]`
> > >
> > >      - GIFT generates more valid samples out-of-distribution, halving the failed generation rate.
> > >
> > > 7) `Model Scale [Fusion360]`
> > >
> > >       - GIFT improves performance with model scale (3b -> 7b), providing evidence of generalization out-of-distribution.
> > >
> > > From these results, we can see that **GIFT is effective out-of-distribution and helps improving the robustness and generation diversity of the base model**.
> > >
> > > -------

---

### Official Review · Reviewer_cgSU · 2026-02-27

**Soundness:** 2
**Presentation:** 3
**Significance:** 3
**Originality:** 2
**Overall Recommendation:** 3
**Confidence:** 5

**Summary:**

This paper proposes GIFT, a self-training framework for image-to-CAD program synthesis that leverages inference-time sampling and geometric verification to improve visual-to-code alignment without online reinforcement learning. By retaining diverse high-IoU programs (Soft-Rejection Sampling) and converting near-miss failures into corrective training examples (Failure-Driven Augmentation), GIFT augments the training distribution to enhance diversity and robustness. Experiments show improved single-shot accuracy, reduced reliance on test-time sampling, and inference cost savings compared to standard supervised baselines.

**Compliance With Llm Reviewing Policy:**

Affirmed.

**Final Justification:**

Replying to Authors‘ Comment

-------

Thank you again for the authors’ efforts.

**On PLAD and GIFT.**
At the level of methodological objectives, after the authors narrowed their claims (framing GIFT as purely a data augmentation method), the core goals of PLAD and GIFT indeed differ: the former is a full training framework, while the latter focuses on data augmentation. Regarding Execution Feedback, there is a factual inaccuracy in the authors’ statement. As stated in the original PLAD paper: *"the P_BEST entry is updated to keep the program whose execution obtains the highest similarity to the input shape."* PLAD uses beam search to sample candidate programs, and relies on executing these programs with a non-differentiable executor and geometric similarity metrics to select the best programs. Therefore, the core idea of data augmentation within the PLAD framework is highly related to the authors’ method, and this warrants a more formal discussion as an important piece of related work. The statement *"ignores inference-time scaling"* is overstated, and the claim that PLAD "ignores diversity" is also inconsistent with PLAD's statement (*"Mixing updates from multiple PLAD methods (LEST+ST+WS) is beneficial because, in this joint paradigm, each method can cover the other's weaknesses."*).

**On generalization.**
Thank you for the additional experiments. Although the improvement is limited compared to the in-distribution setting, it still outperforms the model trained with SFT alone, which partially alleviates my concerns.

**Additional suggestions:**

1. Include comparisons with on-policy RL methods, such as GRPO or more stable variants like GSPO. As they also rely on additional sampling and feedback signals to improve model performance, and thus would provide a more comprehensive set of insights.

2. Provide a more formal discussion of the relationship between GIFT and PLAD in the related work section.

Nevertheless, I am willing to increase my rating.

**Key Questions For Authors:**

1. Since CAD-Coder (Doris et al., 2025) uses the original DeepCAD dataset, which contains many simple and repetitive samples as well as potential train-test leakage, have you evaluated your method after removing duplicated and overlapping samples to verify that SRS and FDA do not amplify these existing biases?
2. Would GIFT achieve better performance than standard RL methods?
3. Considering the potential distribution gap between curated datasets and real-world images, evaluation on the Fusion 360 dataset would help clarify the method’s generalization capability.

**Limitations:**

yes

**Strengths And Weaknesses:**

## **Strengths**
### **1. Valuable and Underexplored Direction**

The paper investigates leveraging inference-time sampling to improve image-to-CAD program synthesis, which has not been sufficiently explored in this domain. Given the inherent one-to-many relationship between CAD geometry and program implementations, and the fact that existing datasets typically provide only one-to-one supervision, exploring this direction is well-motivated and meaningful.

### **2. Effective Strategies with Clear In-distribution Gains**

The proposed SRS and FDA mechanisms address the limitations of rigid SFT by expanding valid solution coverage and targeting near-miss failures. Experimental results demonstrate consistent improvements over the SFT baseline on the GenCAD benchmark under in-distribution settings.

## **Weaknesses**
### **1. Missing Discussion of Closely Related Bootstrapped Program Inference Methods**

The paper does not properly position itself within prior work on bootstrapped program inference. In particular, the proposed executor-based self-training procedure is highly related to PLAD [d], which formalizes a search-train bootstrapping framework for shape program inference using pseudo-labels and approximate distributions. CAD programs are naturally a subcategory of visual programs, making PLAD directly relevant.

However, the manuscript does not cite PLAD or related works in this lineage [a-e], nor does it discuss how the proposed method differs conceptually or algorithmically from these prior approaches. Given the strong structural similarity (model-generated programs, executor validation, construction of pseudo supervision, and subsequent maximum-likelihood updates), this omission makes it difficult to assess what is fundamentally new versus what constitutes an adaptation or engineering extension of an existing framework.

### **2. Poor Writing Quality and Unconvincing Motivation**

The paper suffers from weak writing clarity and logical organization.

In the abstract, several statements are redundant or poorly motivated:
- The opening sentence (***Mapping images to executable CAD programs is a central challenge… yet aligning visual inputs with symbolic code remains difficult***) repeats essentially the same idea without adding precision.
- The claim that ***existing approaches rely on brittle supervised fine-tuning (SFT) or costly reinforcement learning (RL) to overcome data limitations*** is not well supported. In practice, SFT is not merely a workaround for scarce data; rather, domain-specific supervision is crucial because general-purpose pretrained LLMs perform poorly in CAD-specific reasoning tasks [f, g].

The Introduction also has structural issues:
- A substantial portion of the early paragraphs focuses on arguing for code-based CAD representations due to editability and compactness. However, whether the model outputs a serialized sequence (as in many earlier works such as DeepCAD [h]) or human-readable code (e.g., CadQuery) does not fundamentally alter the methodological framework. The choice of representation appears largely orthogonal to the proposed algorithm, yet it receives extensive emphasis in the introduction.
- The authors attribute the motivation of GIFT to weak generalization of SFT (***weak alignment between visual features and program syntax***) and to the inefficiency and instability of RL (***notoriously resource intensive and fragile***). However, based on practical experience, recent RL algorithms such as DAPO [i] and GSPO [j] can operate stably in the CAD domain. Moreover, CAD kernels such as OpenCASCADE generally behave robustly even under intensive computation. The paper presents these issues as primary motivations for GIFT, but does not empirically justify them. Is GIFT truly superior to directly applying RL in terms of resource consumption and model performance? The experimental section does not provide sufficiently comprehensive comparisons with recent RL-based CAD methods [j–m] (or standard modern RL algorithms), leaving the claimed advantages only partially supported.

### **3. Weak OOD Generalization**

The method shows a significant performance drop on real-world OOD images (Figure 15, Appendix), raising concerns about its generalization ability. The FDA strategy may further encourage overfitting, as it augments training data using near-miss samples from the same distribution. Moreover, the paper does not evaluate on additional external datasets (e.g., Fusion 360), making it difficult to assess robustness beyond the training domain.

### **4. Incomplete Baseline Comparisons**

The empirical evaluation is not comprehensive. The paper does not provide a thorough comparison against recent RL-based CAD generation methods [j–m] (at least including standard modern RL algorithms such as GRPO and GSPO) or relevant image-to-CAD approaches [l, n]. Although Table 4 reports related comparison results, some metrics are directly cited from the original papers, and due to differences in experimental settings and evaluation protocols, such direct comparisons may not be entirely fair. For example, methods such as cadrille [l] compute IoU based on point clouds, while others such as [k], although reporting the same $IoU_{best}$ metric as this paper, evaluate on a cleaned dataset with simple and repetitive models removed, making it difficult to draw clear and controlled conclusions about the relative performance of the proposed method.


----

[a] Geoffrey E Hinton, Peter Dayan, Brendan J Frey, and Radford M Neal. The “wake-sleep” algorithm for unsupervised neural networks. Science, 268(5214):1158–1161, 1995.

[b] Kevin Ellis, Catherine Wong, Maxwell Nye, Mathias Sable Meyer, Luc Cary, Lucas Morales, Luke Hewitt, Armando Solar-Lezama, and Joshua B. Tenenbaum. Dreamcoder: Growing generalizable, interpretable knowledge with wake-sleep bayesian program learning, 2020.

[c] Luke B. Hewitt, Tuan Anh Le, and Joshua B. Tenenbaum. Learning to learn generative programs with memoised wake sleep, 2020.

[d] Jones, R. Kenny et al. “PLAD: Learning to Infer Shape Programs with Pseudo-Labels and Approximate Distributions.” 2022 IEEE/CVF Conference on Computer Vision and Pattern Recognition (CVPR) (2020): 9861-9870.

[e] Ganeshan, Aditya, R. Kenny Jones, and Daniel Ritchie. "Improving unsupervised visual program inference with code rewriting families." Proceedings of the IEEE/CVF International Conference on Computer Vision. 2023.

[f] Wang, Ruiyu, et al. "Text-to-cad generation through infusing visual feedback in large language models." arXiv preprint arXiv:2501.19054 (2025).

[g] Li, Jiahao et al. “CAD-Llama: Leveraging Large Language Models for Computer-Aided Design Parametric 3D Model Generation.” 2025 IEEE/CVF Conference on Computer Vision and Pattern Recognition (CVPR) (2025): 18563-18573.

[h] Wu, Rundi et al. “DeepCAD: A Deep Generative Network for Computer-Aided Design Models.” 2021 IEEE/CVF International Conference on Computer Vision (ICCV) (2021): 6752-6762.

[i] Yu, Qiying et al. “DAPO: An Open-Source LLM Reinforcement Learning System at Scale.” ArXiv abs/2503.14476 (2025): n. pag.

[j] Zheng, Chujie et al. “Group Sequence Policy Optimization.” ArXiv abs/2507.18071 (2025): n. pag.

[k] Li, Jiahao et al. “ReCAD: Reinforcement Learning Enhanced Parametric CAD Model Generation with Vision-Language Models.” ArXiv abs/2512.06328 (2025): n. pag.

[l] Kolodiazhnyi, Maksim et al. “cadrille: Multi-modal CAD Reconstruction with Reinforcement Learning.” (2025).

[m] Niu, Ke, et al. "From Intent to Execution: Multimodal Chain-of-Thought Reinforcement Learning for Precise CAD Code Generation." arXiv preprint arXiv:2508.10118 (2025).

[n] Wang, Siyu et al. “CAD-GPT: Synthesising CAD Construction Sequence with Spatial Reasoning-Enhanced Multimodal LLMs.” ArXiv abs/2412.19663 (2024): n. pag.

---

> ### Author Rebuttal · Authors · 2026-03-26
>
> Thank you for the detailed review. Several of your concerns point to places where our positioning was unclear, and we have revised the paper to make these points explicit.
>
> ### ***GIFT is a Data Augmentation Mechanism***
>
> GIFT is fundamentally a **data augmentation mechanism**, not a new model or training algorithm. It leverages **inference-time scaling** and **geometric kernel feedback** to build an **augmentation scheme specifically tailored to CAD generation**. Because GIFT outputs a dataset, it integrates with any post-training pipeline. We used standard SFT as a proof of concept, showing that even basic post-training is highly effective when fueled by our systematic data exploration methodology.
>
> -------
>
> ### **Novelty and Prior Work**
>
> **Concern**: Failure to cite PLAD/Wake-Sleep
>
> **Authors**:  Thank you for pointing us to these relevant papers. We agree GIFT is closely related to prior search-train / pseudo-labeling frameworks such as PLAD and wake-sleep-style bootstrapping, and we have added those citations. Our intended novelty claim is narrower: in image-to-CAD, we use kernel-verified near-miss programs not only as additional valid pseudo-labels (SRS) but also as rendered corrective examples (FDA). We have revised the abstract and introduction to position GIFT as a CAD-specific extension of that lineage rather than as a wholly separate paradigm
>
> -------
>
> ### **Motivation: SFT, RL, and Data Scarcity**
>
> **Concern**: SFT is not just a workaround for scarce data; RL algorithms can operate stably.
>
> **Authors**: We completely agree that SFT is crucial for CAD tasks, not merely a workaround, and have updated our abstract and introduction to reflect this. As shown mathematically in Appendix G (Eq. 12), SFT provides essential grounding, while our SRS and FDA gradients supply diversity and robustness. Our use of the term "brittle" referred to the limitations of small datasets, not a flaw in the SFT algorithm.
>
> Our intended claim is not that SFT is flawed or that RL is inherently unstable. Rather, for this setting, we found the lack of diverse, well-aligned image-program pairs to be a major bottleneck, and GIFT addresses that bottleneck by converting inference-time exploration into additional supervised training data.
>
> ------
>
> ### **Evaluation**
>
> **Concern**: Evaluation is not comprehensive against recent RL/CAD methods.
>
> **Authors:** One source of mismatch across reported numbers is whether execution / B-Rep failures are counted. To avoid overstating the comparison, we now clarify this protocol difference explicitly. We do not claim the table is a definitive apples-to-apples head-to-head against cadrille; rather, it illustrates that counting failures materially changes the gap between a SOTA multimodal model and a simple SFT model tuned with GIFT drops significantly.
>
> | Name  | IoU (mean) | IoU (median) | Notes    |
> | ----------- | ---------- | ------------ | ---------------|
> | cadrille w/o f  | 0.92       | 0.98  | Excludes generation failures |
> | cadrille w/ f    | 0.83       | 0.97  | Includes generation failures |
> | CAD-Coder-SFT w/ f        | 0.70 | 0.80  | Baseline SFT |
> | CAD-Coder-GIFT w/ f       | 0.78 | 0.95 | GIFT Augmented (Ours) |
> | Relative Delta (cadrille w/ f, GIFT w/ f) | -0.06   | -0.02        | Median Gap drastically reduced   |
>
>
> ------------
>
> ### **Alignment & Robustness to Problem Complexity**
>
>
> **Concern**: Does the method truly align visual features with program syntax?
>
> **Authors:** Intrinsic text-generation metrics provide supporting evidence of better image-to-code alignment, though we do not view them as a complete substitute for geometric evaluation. GIFT improves BLEU by 8.7% and METEOR by 4.0% over the SFT baseline. GIFT generates clean, human-readable syntax rather than just exploiting topological loopholes. It solves 53.8% more problems overall (Fig. 2/16/17), maintaining high geometric coherence even as design complexity increases.
>
> -------
>
> ### **Dataset Integrity and OOD**
>
> **Concern**: Potential train-test leakage in DeepCAD; OOD performance drops
>
> **Authors:** Under an exact-match deduplication check, we found no exact train-test duplicates. We cannot rule out near-duplicates, so we will clarify that limitation. Additionally, our SRS actively counteracts mode collapse by rejecting exact code matches (IoU=1) and retaining diverse valid alternatives, forcing a broader distribution.
>
> OOD: We agree that evaluation on Fusion 360 would strengthen the paper. We do not have that experiment in this submission, and we will present Fig. 15 only as a relative stress test showing that GIFT remains better than SFT under the same OOD setting.
>
> --------
>
> ### **Manuscript Revisions**
>
> **Authors:** We have rewritten the abstract and introduction. Instead of discussing SFT or RL, the revised text clearly emphasizes that the primary bottleneck in generative CAD design is the scarcity of diverse training examples that successfully align visual geometry with program syntax.
>
> ------

---

> > ### Author Rebuttal · Reviewer_cgSU · 2026-04-01
> >
> > Thank you for your efforts. I appreciate the clarifications and the revisions made after the initial review. After reading the rebuttal, however, I believe my main concerns are only partially resolved.
> >
> > **On Positioning and Core Claims.** The rebuttal clarifies that GIFT is *"fundamentally a data augmentation mechanism,"* yet the paper frames it as a complete fine-tuning framework and positions it as an alternative to online RL. This significantly overstates the novelty relative to prior work, especially PLAD [d]. Both papers share the same primary motivation: criticizing RL as the main foil. PLAD states *"RL is notoriously unstable and slow to converge,"* and GIFT echoes *"Reinforcement Learning (RL) can improve alignment but is notoriously resource-intensive and fragile ... executing CPU-bound CAD kernels during online training."* This criticism was reasonable in 2021, but in 2026 algorithms such as GRPO, DAPO, and GSPO have demonstrated stable RL training at scale for LLM training (even in domain-specific setting), and the paper does not engage with this development. On the method side, SRS is conceptually equivalent to PLAD's Self-Training with an additional IoU threshold window. The one genuinely new element is FDA. We encourage the authors to reposition GIFT within the existing literature and clearly articulate what is fundamentally new relative to prior work.
> >
> > **About Generalization.** FDA has a potential overfitting problem. Moreover, the authors do not provide generalization results on datasets such as Fusion360. The current evidence cannot support the claim that *"GIFT-trained models achieve superior generalization"* According to the current literature, the only work that provides generalization results of CAD-Coder-SFT (the backbone used in this paper) on the Fusion360 dataset is ReCAD [k]. Their results show that the original CAD-Coder-SFT has weak generalization. Although the proposed method shows better performance in-distribution, the GenCAD dataset is derived from DeepCAD, which contains a large number (tens of thousands) of very simple models (e.g., cuboids and cylinders), and also contains many duplicated samples.
> >
> > > It is recommended to perform deduplication using image embeddings, and you can find many train-test leakage samples, for example: [test 00017091, train 00017077], [test 00020602, train 00021050], [test 00036066, train 00238749]
> >
> > This can introduce potential bias into the experimental results. Without stronger cross-dataset evaluation, it is difficult to know whether FDA genuinely improves robustness or instead further amplifies pre-existing dataset biases. For that reason, I do not think this is merely a limitation to be acknowledged at the end of the paper; rather, it is an issue that should be addressed in the experimental design stage. I suggest adding external generalization experiments (e.g., using Fusion 360 or CAD-MLLM).
> >
> > **About Table 4.** My concern regarding Table 4 is not resolved by the rebuttal. I understand the authors’ clarification that one source of mismatch is whether failures are counted, but because most existing methods have relatively low failure rates, this clarification alone does not make the comparison sufficiently fair or informative. The more fundamental issue is that the compared methods do not share a unified dataset and evaluation pipeline. In this problem, the metrics are highly sensitive to implementation details, and different IoU protocols can produce substantially different conclusions. Because the paper makes repeated claims about online RL, a more convincing empirical case would require a fairer and more comprehensive comparison with RL methods under a unified setting.
> >
> > For these reasons, I believe the paper would require a major revision, and I therefore keep my original score unchanged.

---

> > > ### Author Response · Authors · 2026-04-04
> > >
> > > ## Ablation Experiments
> > >
> > > ## https://anonymous.4open.science/r/gift-rebuttal-F818/README.md
> > >
> > > --------
> > >
> > > **Authors**:
> > >
> > > Thank you for your constructive feedback. We have addressed your concerns below and conducted new `Fusion360` experiments (`Ablation 1 to 7` in the anonymized repository). Please refer to Reviewer `QNNZ` response for details.
> > >
> > > We hope these ablations can resolve your major concerns regarding our work.
> > >
> > > -------
> > >
> > > **Reviewer**:
> > >
> > > > The rebuttal clarifies that GIFT is "fundamentally a data augmentation mechanism," yet the paper frames it as a complete fine-tuning framework and positions it as an alternative to online RL.
> > >
> > > **Authors**:
> > >
> > > Our abstract explicitly states: "In this work, we ask: **how far can we push performance by leveraging test-time compute to bootstrap an augmented training set**?". This is the core framing of our work. We have revised the abstract and introduction to ensure this data-augmentation framing is clear.
> > >
> > > -----
> > >
> > > **Reviewer:**
> > >
> > > > This significantly overstates the novelty relative to prior work, especially PLAD [d].
> > >
> > > **Authors:**
> > >
> > > We respectfully disagree. **PLAD and GIFT fundamentally differ in their core objectives, architectures, inference, and use of execution feedback**:
> > >
> > > ### `Method`
> > >
> > > - PLAD is a 2D/3D shape reconstruction method using a separete recognition model $p(z|x)$ that requires shape conditioning. Technically, the recognition model is not part of the generative model. Some PLAD variants are deterministic $x = d(z)$, others are generative $p(x | z) p(z)$ with an unconditional prior.
> > >
> > > - **GIFT is a data augmentation method for image-to-code CAD models**. GIFT assumes access to a conditional generative model $p(x|z) p(z|c)$. **GIFT does not require a separate recognition model**.
> > >
> > > ### `Execution Feedback`
> > >
> > > - PLAD relies on sampling-based beam search to generate code without geometric or execution feedback, using the executor only later to convert programs to shapes.
> > >
> > > - None of PLAD's variants (ST, LEST, WS) use SRS or FDA augmentations, meaning **PLAD code generation and selection are unaware of feedback** and can produce ungrounded tuples or duplicate code.
> > >
> > >
> > > - **GIFT centers its entire test-time strategy around execution feedback**, using intersection over union (IoU) metrics to drive SRS and FDA augmentations.
> > >
> > > - GIFT explicitly filters out exact duplicates and **modifies one modality while anchoring the other to the underlying training data**.
> > >
> > > ### `Inference-Time Scaling`
> > >
> > > - PLAD treats generation as training-time computation and ignores inference-time scaling, code quality, and diversity.
> > >
> > > - GIFT explicitly leverages modern language model properties to perform inference-time scaling for augmentation, selection, and tuning.
> > >
> > > ### `Novelty`
> > >
> > > GIFT’s core focus is **leveraging test-time compute to bootstrap an augmented training set**, whereas PLAD is shape reconstruction. Following your feedback, we have improved the introduction framing.
> > >
> > >
> > > **We commit to include PLAD-LEST as a baseline in the camera-ready version**.
> > >
> > > -----
> > >
> > > **Reviewer**:
> > >
> > > > FDA has a potential overfitting problem.
> > >
> > > **Authors**:
> > >
> > > FDA acts like classic image augmentation (e.g., random cropping) to improve robustness. However, rather than applying random changes, FDA leverages inference-time compute to actively select augmentations based on the specific task, model scaling behavior, and geometric kernel feedback.
> > >
> > > ------
> > >
> > > **Reviewer**:
> > >
> > > > It is recommended to perform deduplication using image embeddings,
> > >
> > > **Authors**:
> > >
> > > Because a 2D image is a lossy approximation of a 3D shape, similar images do not inherently produce similar code. We verified this using a `jina-code-embedding` model and syntactic matching, finding zero exact (image, code) duplicates. When possible, we think it's better to leave the test set unmodified to avoid introducing unintended evaluation bias.
> > >
> > >
> > > ---------
> > >
> > > **Reviewer**:
> > >
> > > > relatively low failure rates
> > >
> > > **Authors**:
> > >
> > > **Failures cluster around the hardest problems** (`Fig 2b/2c`), reaching higher % in out-of-distribution scenarios. Excluding them biases results toward easier tasks, so including all failures actually lower-bounds GIFT's performance. Furthermore, because GIFT prioritizes distribution robustness and diversity over simply optimizing IoU, `Figure 2` emphasizes intrinsic metrics like output quality, solve ratios, and complexity.
> > >
> > > ----
> > >
> > > **Reviewer**:
> > >
> > > > the compared methods do not share a unified dataset and evaluation pipeline.
> > >
> > > **Authors**:
> > >
> > > This is an intrinsic challenge in CAD generation, as CAD models use different DeepCAD splits and modalities. Our goal is to contextualize our results within the broader community. Our failure/no-failure and median-based analyses demonstrate how to achieve robust, fair evaluations despite these pipeline variations.
> > >
> > > See `IoU Ablation 8) and 9)` in the anonymized repo for IoU pre-processing comparisons. We show that **removing the "best" orientation match changes the in-distribution results by <1% at any budget**.

---

### Official Review · Reviewer_YyQC · 2026-03-12

**Soundness:** 3
**Presentation:** 3
**Significance:** 3
**Originality:** 2
**Overall Recommendation:** 4
**Confidence:** 3

**Summary:**

At a high level, the paper aims to improve image-to-CAD generative models. The authors argue that a major limitation of current CAD generative models is that, during training, they are supervised using only a single ground-truth program per image, even though multiple CAD programs can produce the same shape.
To address this, they propose the GIFT method. GIFT uses geometric verification at inference time on a base model to create a higher-quality training dataset, which is then used to fine-tune the base model and obtain the final model.
More specifically, the method first generates diverse CAD programs for a given image using the base model with different temperature settings. For each generated program, an alignment score with respect to the ground truth is computed using a geometric validity function based on a CAD kernel. This alignment score, measured using IoU, is then used to construct the fine-tuning dataset.
If the alignment score is below 0.5, the corresponding shapes are removed from the training pool. If the score is above 0.9, they are retained in the training set through what the authors call Soft Rejection Sampling. If the alignment score falls between 0.5 and 0.9, the method applies Failure-Driven Augmentation, where new datapoints are added by rendering the CAD program that is geometrically close to the target shape and including it in the training set.
Finally, the model is trained on the dataset obtained from the supervised data, Soft Rejection Sampling, and Failure-Driven Augmentation. The paper then presents extensive ablation studies and comparisons against multiple baselines.

**Compliance With Llm Reviewing Policy:**

Affirmed.

**Key Questions For Authors:**

Can you please discuss the heavy reliance on thresholds and heuristics, as well as the major weaknesses of FDA? Moreover, it seems to me that this method only works if the base model is already very strong. Could you discuss that as well?

**Limitations:**

yes

**Strengths And Weaknesses:**

Strengths:

The paper addresses an important but relatively underexplored topic: CAD shape generation. In addition, it explores test-time methods in this domain, which is interesting and appears to yield strong results.

I found the paper to be well written, with clear figures that explain the method in detail and make it easy to understand.

The paper also seems technically sound and reasonably reproducible. Overall, the topic is relevant, and the proposed approach appears meaningful for improving results.

Weaknesses:

Several major related works are missing, including: (https://arxiv.org/pdf/2512.03018, https://arxiv.org/pdf/2401.15563 and https://arxiv.org/pdf/2203.13944)

Failure-Driven Augmentation (FDA) appears to have a significant weakness because it introduces ambiguous supervision: the model is trained on both ground-truth image-to-CAD pairs and failure-rendered image-to-CAD pairs.
I also think the method relies heavily on thresholds and heuristics, which may not generalize well across shapes of varying complexity and different datasets.

Although the approach avoids multiple inferences at test time, it still requires substantial dataset creation during the fine-tuning stage.

---

> ### Author Rebuttal · Authors · 2026-03-26
>
> **Reviewer**:
>
> > The paper addresses an important but relatively underexplored topic: CAD shape generation. In addition, it explores test-time methods in this domain, which is interesting and appears to yield strong results.
>
> **Authors**:
>
> Thank you for highlighting the significance of this domain and the effectiveness of our approach. We agree that image-conditional CAD generation remains a central challenge in generative design. By amortizing expensive inference-time search directly into the model parameters , GIFT successfully captures the performance benefits of test-time scaling while remaining computationally efficient during deployment.
>
> ------
>
> **Reviewer**:
>
> > I found the paper to be well written, with clear figures that explain the method in detail and make it easy to understand.
>
> **Authors**:
>
> We sincerely appreciate your positive feedback on the clarity and presentation of our work. Our goal was to clearly illustrate how offline geometric verification can systematically convert inference-time samples into a high-quality training set. We are glad the figures effectively communicated the dual mechanisms of Soft-Rejection Sampling and Failure-Driven Augmentation.
>
> ------
>
> **Reviewer**:
>
> > The paper also seems technically sound and reasonably reproducible. Overall, the topic is relevant, and the proposed approach appears meaningful for improving results.
>
> **Authors**:
>
> Thank you for recognizing the technical soundness and reproducibility of our framework. We explicitly designed GIFT to replace an online reward-optimization loop with an offline data-generation stage followed by standard supervised fine-tuning. By separating exploration (offline verification) from learning (standard supervised updates), we ensure the method is not only robust and straightforward to reproduce, but also highly effective at improving generation quality.
>
> --------
>
> **Reviewer**:
>
> > related works are missing
>
> **Authors**:
>
> Thank you for pointing this line of work out. We have included them in the related work.
>
> -------
>
> **Reviewer**:
>
> > FDA introduces ambiguous supervision: the model is trained on both ground-truth image-to-CAD pairs and failure-rendered image-to-CAD pairs.
>
> **Authors**:
>
> - Supervision: FDA acts as a targeted geometric denoising objective. By rendering near-miss programs into synthetic images and pairing them with the original ground-truth code, the model learns to recover from its own structural errors. SRS and FDA are applied on disjoint IoU intervals and serve complementary roles: SRS broadens the set of valid solutions, while FDA teaches recovery from structured near-miss errors.
>
> - Robustness: Fig 2/16/17 demonstrate that GIFT maintains significantly higher resilience than the SFT baseline as task complexity (measured by token length and operations) increases. This confirms that learning from targeted near-miss failures actually helps the model sustain geometric coherence even for intricate designs.
>
> -------
>
> **Reviewer**:
>
> > it still requires substantial dataset creation during the fine-tuning stage.
>
> **Authors**:
>
> We agree GIFT shifts compute to an offline data-generation stage. We will present this explicitly as a train-time/test-time trade-off, not as free efficiency. The cost is incurred once per base model / training set, after which the resulting augmented data can support cheaper single-shot deployment.
>
> ---
>
> **Reviewer**:
>
> > Can you please discuss the heavy reliance on thresholds and heuristics
>
> **Authors**:
>
> As we extensively analyze in Section 3.1 (Threshold selection) and Section 4.1 (Inference-Time Scaling: Analysis), every threshold we use is a function of the scaling behavior of the model on the underlying problem. The goal of GIFT is to augment the training distribution is such a way to leverage the strengths and address the weaknesses of the model for that specific task (image-to-code CAD generation). This is achieved trough geometric feedback and scaling compute at inference time.
>
> We agree the method uses thresholds. Our point is that these **thresholds are chosen from the model’s empirical scaling behavior on the task** rather than hand-picked per example. We will make this distinction clearer and temper any implication that the chosen values are universally optimal.
>
> ------
>
> **Reviewer**:
>
> > this method only works if the base model is already very strong.
>
> **Authors**:
>
> GIFT does assume a base model that can produce a non-trivial fraction of executable or near-miss samples. It does not require a perfect sampler - our current base model still produces 40% samples below 0.9 IoU - but we will make this assumption explicit rather than implying GIFT is model-agnostic.

---

> > ### Author Rebuttal · Reviewer_YyQC · 2026-04-04
> >
> > I appreciate the authors’ response. i would like to keep this as my current rating.

---

> > > ### Author Response · Authors · 2026-04-04
> > >
> > > ## Ablation Experiments
> > >
> > > ## https://anonymous.4open.science/r/gift-rebuttal-F818/README.md
> > >
> > > -------
> > >
> > > Thank you for the feedback and discussion.
> > >
> > > Please see also the response to Reviewer `QNNZ`, where we provide additional evaluation for `fusion360`, an out-of-distribution CAD dataset. Results are available in the anonymized repository.

---

### Official Review · Reviewer_xLhT · 2026-03-12

**Soundness:** 3
**Presentation:** 3
**Significance:** 3
**Originality:** 3
**Overall Recommendation:** 4
**Confidence:** 3

**Summary:**

This paper introduces Geometric Inference Feedback Tuning (GIFT), an offline self-training framework designed to improve the synthesis of parametric CAD programs from 2D images. A way to counter online Reinforcement Learning (RL) with rendering based rewards being computationally prohibitive due to CPU-bound geometric kernels. The authors instead generates rollouts offline and use a deterministic CAD kernel (OpenCASCADE) to verify them and filter them into two augmentation datasets: Soft-Rejection Sampling (SRS), which preserves diverse, high-fidelity programs (0.9 ≤ IoU < 0.99) to soften the training distribution, and Failure-Driven Augmentation (FDA), which renders near-miss failures (0.5 ≤ IoU < 0.9) as noisy augmentationss. By training on this combined dataset, GIFT amortizes expensive inference-time search into the model weights, demonstrating a 12% improvement in single-shot Pass@1 accuracy over the SFT baseline and matching the performance of extensive test-time scaling with an 80% reduction in inference compute.

**Compliance With Llm Reviewing Policy:**

Affirmed.

**Key Questions For Authors:**

* Can you provide a baseline comparing the FDA strategy to standard 2D image augmentations (e.g., adding noise, blur, or occlusion to the original dataset images)? This would help prove that the geometric structural noise of the FDA renders is uniquely responsible for the performance gains.

**Limitations:**

yes

**Strengths And Weaknesses:**

Strengths:
* By decoupling the geometric verification from the training loop, GIFT elegantly avoids the severe computational overhead of running CPU-bound CAD solvers during online RL, making high-quality CAD synthesis much more scalable.
* The paper demonstrates compelling performance gains, improving the Pass@1 mean IoU from 0.698 to 0.782.

Weaknesses:
* The theoretical premise of FDA is risky. By training the model to output the code for Object B (the ground truth) when conditioned on an image of Object A (the rendered near-miss), the model is effectively being taught to ignore discrepancies between the visual input and the target geometry. This "denoising" risks inducing systematic hallucinations, where the model might confidently output familiar CAD structures even when the input image exhibits fine-grained differences.
* It is unclear if the performance boost from FDA is truly due to the geometric nature of the rendered failures, or simply due to the addition of visual noise. A baseline comparing FDA against standard 2D image augmentations (e.g., blur, crop, color jitter) applied to the original images is missing.

---

> ### Author Rebuttal · Authors · 2026-03-26
>
> **Reviewer**:
>
> > GIFT elegantly avoids the severe computational overhead of running CPU-bound CAD solvers during online RL, making high-quality CAD synthesis more scalable.
>
> **Authors**:
>
> Thank you for highlighting this architectural advantage. As you noted, executing CPU-bound CAD kernels during online RL creates a severe computational bottleneck. By decoupling exploration from training and moving verification offline, GIFT preserves the rich signal of geometric feedback while avoiding the need to place CAD-kernel execution inside the training loop. This design is what allows us to capture the performance benefits of test-time scaling while simultaneously reducing inference compute by 80%.
>
> ---------
>
> **Reviewer**:
>
> > The paper demonstrates compelling performance gains, improving the Pass@1 mean IoU from 0.698 to 0.782.
>
> **Authors**:
>
> Thank you for recognizing these results. This 12% improvement in mean IoU validates the effectiveness of our dual augmentation strategy. By combining Soft-Rejection Sampling (for diversity) and Failure-Driven Augmentation (for robustness), GIFT effectively amortizes the cost of expensive inference-time search directly into the model's weights using a stable, simple supervised objective. This dramatically improves single-shot generation and reduces the amortization gap between Pass@1 and Pass@k by over 66% compared to the baseline.
>
> ----------
>
> **Reviewer**:
>
> > This "denoising" risks inducing systematic hallucinations, where the model might confidently output familiar CAD structures even when the input image exhibits fine-grained differences.
>
> **Authors**:
>
> We agree FDA could introduce ambiguity if the rendered failure were too far from the target. **Our claim is narrower: by restricting FDA to kernel-verified near-miss programs, we find it improves robustness without evidence of increased hallucination in our current analyses**. Unlike standard heuristic augmentations like random noise, FDA leverages a CAD kernel to identify the model's specific near-miss errors. By rendering these exact geometric failures back into images and pairing them with ground-truth code, FDA acts as a targeted geometric denoising objective. In practice, this gives the model corrective supervision on the specific structural errors it already makes, rather than only exposing it to generic image noise.
>
> **As demonstrated by our intrinsic metrics and robustness analysis (Figures 2 and 16), GIFT maintains significantly greater resilience on more complex, high-token designs**. This empirically confirms that the model learns to resolve complex geometric details without collapsing into systematic hallucinations.
>
> --------
>
> **Reviewer**:
>
> > A baseline comparing FDA against standard 2D image augmentations
>
> **Authors**:
>
> Thank you for this suggestion. To address it, we have extended the ablation that applies standard stochastic 2D image augmentations during tuning. The pipeline includes random flips, color jittering, random erasing, and affine transformations. While these augmentations encourage generalization across viewpoint and lighting variations, they are heuristic and lack geometric validation.
>
> | Augmentation        | Parameter       | Value              |
> |--------------------|---------------|--------------------|
> | Horizontal Flip        | Probability       | 0.5                |
> | Vertical Flip             | Probability      | 0.5                |
> | Color Jitter              | Brightness      | ±0.2               |
> |                                  | Contrast         | ±0.2               |
> |                                  | Saturation      | ±0.2               |
> |                                  | Hue                 | ±0.1               |
> | Random Erasing      | Probability     | 0.5                |
> |                                   | Scale           | 2%–33%             |
> |                                   | Aspect Ratio    | 0.3–3.3            |
> | Random Affine         | Rotation        | ±45°               |
> |                     | Translation     | ±20%               |
> |                     | Scale              | 0.75×–1.25×     |
>
> As shown in the table below, applying these standard 2D augmentations to our baseline yields only a marginal improvement in mean IoU (from 69.8 to 70.7). In contrast, our Failure-Driven Augmentation (FDA) strategy leverages the problem structure to independently raise performance to 76.2.
>
> | Method	| IoU (mean) |
> |--------------------|--------------------|
> | SFT	| 69.8 |
> | w/ RANDOM AUG	| 70.7 |
> | w/ FDA AUG	| 76.2 |
>
> Under the same fine-tuning budget, standard 2D augmentations improve mean IoU only from 69.8 to 70.7, whereas FDA reaches 76.2. This suggests the benefit comes from structured geometric perturbations, not generic visual noise alone.
> The structured, geometric feedback provided by FDA - which visually manifests the model's actual topological misinterpretations - drives the significant gains in robustness and performance.

---

### Decision · Program_Chairs · 2026-04-30

**Decision:**

Accept (regular)

**Comment:**

This paper received 3 weak accept and 1 weak reject. The reviewers generally agree that it presents a technically sound and practically useful contribution to image-to-CAD program synthesis, with consistent empirical gains and a compelling data-centric perspective. The main concerns raised by rev. cgSU revolve around the novelty and positioning relative to prior work such as PLAD, as well as evaluation completeness and generalization.

The rebuttal successfully addressed most technical concerns, particularly regarding FDA and augmentation effectiveness, and clarified the scope of the contribution. While the relationship to PLAD remains a point of discussion, it primarily reflects positioning and citation issues rather than a fundamental flaw.

Overall, the strengths, especially the empirical performance, practical framework, and novel use of geometric feedback for augmentation, outweigh the remaining concerns. This meta-review thus  recommends weak accept.  The paper would benefit from improved positioning and additional discussion of related work in the final version.